# Less Is More: Fast and Accurate Reasoning with Cross-Head Unified Sparse Attention

## Abstract

Large reasoning models achieve strong performance through test-time scaling. However, this comes at the cost of substantial computational overhead, particularly from excessive token generation on short input prompts. While sparse attention mechanisms can reduce latency and memory usage, existing approaches suffer from accuracy degradation due to accumulated errors during long-generation reasoning. These methods generally require either high token retention or costly retraining. We introduce LessIsMore, a training-free sparse attention mechanism for reasoning tasks. Unlike existing approaches that rely on head-specific local optimizations, LessIsMore leverages global attention patterns by aggregating token selections across heads with recent contextual information. This unified cross-head ranking enables more efficient token selection for future decoding layers, eliminating the need to maintain separate token subsets per head and improving both generalization and efficiency. Evaluation across diverse reasoning tasks and benchmarks shows that LessIsMore preserves—and in some cases improves—accuracy while achieving up to $1.6\times$ end-to-end decoding speedup compared to full attention. Moreover, LessIsMore attends to $2\times$ fewer tokens without accuracy loss and accelerates sparse attention computation by up to $1.72\times$ compared to existing methods.

## 1 Introduction

Recent advancements in large reasoning models (LRMs) have significantly enhanced the reasoning capabilities of large language models (LLMs). Models such as DeepSeek-R1 (DeepSeek-AI, 2025), Gemini-2.5-pro (DeepMind, 2025), OpenAI-o3 (OpenAI, 2025), Qwen3 (Team, 2025), and gpt-oss (OpenAI, 2025b) demonstrate strong performance by leveraging test-time scaling to improve accuracy on challenging reasoning benchmarks (Wei et al., 2023; AoPS, 2025; Rein et al., 2023).

Unlike traditional language processing tasks, which involve long inputs and short outputs, reasoning tasks exhibit a different computational profile. They require generating extensive multi-step derivations—often spanning tens of thousands of output tokens (Research, 2024)—from relatively concise problem statements. This decode-heavy nature leads to substantial computational overhead (Liu et al., 2025). For example, under full attention in the HuggingFace framework, DeepSeek-R1-Distill-Llama-8B requires more than 20 minutes on an NVIDIA RTX A5000 GPU to produce 32,768 tokens for a single AIME problem.

This paradigm creates a unique optimization opportunity: while input processing benefits from full attention for accurate context understanding, the lengthy generation phase is well-suited to sparse attention mechanisms (Cai et al., 2025; Gao et al., 2025). Sparse attention reduces computational complexity and generation latency by selectively attending to a subset of critical tokens. Current techniques fall into two categories: selection-based methods (Yang et al., 2024; Tang et al., 2024; Hao et al., 2025; Liu et al., 2024; Gao et al., 2025; Yuan et al., 2025), which retain the full key-value (KV) cache but utilize a subset of tokens during attention computation, and eviction-based methods (Li et al., 2024; Xiao et al., 2023; Zhang et al., 2023; Adnan et al., 2024; Cai et al., 2025), which permanently discard tokens deemed unimportant.

However, existing sparse attention approaches suffer from significant accuracy degradation on reasoning tasks due to the accumulation of selection errors across long generation sequences (Gao et al., 2025). Although standard generation tasks can tolerate moderate information loss, step-by-step reasoning requires crucial contextual information to be preserved throughout the entire derivation to

maintain logical consistency (Lee & Hockenmaier, 2025). For instance, TidalDecode (Yang et al., 2024) achieves over 99.9% sparsity with no accuracy loss on retrieval tasks, but must reduce sparsity below 50% to preserve accuracy on AIME-24 reasoning tasks. In these settings, even small selection inaccuracies compound over thousands of generated tokens, leading to attention recall degradation and cascading accuracy drops. Moreover, prior work has shown that inaccurate sparse attention in reasoning models can also prolong generation, further reducing overall inference efficiency (Gao et al., 2025).

These limitations motivated us to investigate the intrinsic attention distributions of reasoning models and tasks in search of patterns that enable more accurate and efficient token selection. Our token-level analysis of the reasoning process reveals two key locality patterns in attention that fundamentally challenge the selection principles used in existing sparse attention methods.

First, reasoning tasks exhibit prominent *spatial locality* across attention heads, particularly in the Grouped Query Attention (GQA) frameworks prevalent in open-source LLMs (Touvron et al., 2023; AI, 2024b; Team, 2025). Contrary to typical wisdom that different heads specialize in distinct roles requiring separate token subsets (Yang et al., 2024; Xiao et al., 2024; Tang et al., 2024), our analysis suggests substantial overlap in token-importance rankings across heads in the same decoding layer. This reveals that per-head top-$k$ selection yields only a local optimum—overfitting to head-specific query patterns while potentially missing globally important tokens that could enhance performance in future layers.

Second, we observe a *recency locality* pattern across decoding steps: recently generated tokens consistently receive higher attention in subsequent steps. Notably, the ratio between the size of this "recency window" and the total number of selected tokens remains relatively constant throughout decoding. This reflects the intuition that each logical step in reasoning builds directly on the conclusions of preceding steps (Lee & Hockenmaier, 2025).

Building on these insights, we present LessIsMore, a novel *training-free* sparse attention approach that achieves *higher* accuracy on reasoning tasks with *lower* latency, enabled by a kernel-friendly design that attends to *fewer* tokens. LessIsMore employs *Cross-Head Unified Sparse Attention*, which combines head-specific local information with a global attention pattern to enable more robust and accurate token selection. In each selection layer, LessIsMore leverages the identified locality patterns through a unified process: each attention head first selects its approximate top-$k$ tokens using tailored selection schemes; these candidate tokens are then aggregated across heads, globally ranked, and pruned to satisfy a predefined token budget. To preserve recency locality, LessIsMore further reserves a fixed portion of this for a stable recency window, ensuring that recently generated tokens—critical for step-by-step reasoning—are consistently attended to.

Evaluation on DeepSeek-R1-Distill-Llama-8B (DeepSeek-AI, 2025) and Qwen3 models (4B, 8B, and 14B) (Team, 2025) across diverse reasoning tasks, including AIME-24/25, GPQA-Diamond, and MATH500, demonstrates that LessIsMore consistently and significantly outperforms existing sparse attention baselines, including reasoning-focused methods that require retraining. LessIsMore preserves full accuracy at substantially higher sparsity levels—achieving up to 87.5% sparsity on AIME-24 with lossless accuracy—while avoiding any increase in reasoning length. These improvements are further enabled by our customized kernel optimizations for GQA models. In terms of efficiency, LessIsMore achieves up to $1.6\times$ end-to-end per-token decoding speedup compared to full attention.

In summary, our contributions are:

- We present the first detailed, token-level analysis of attention distributions in reasoning tasks, revealing fundamental *spatial* and *recency* locality patterns that challenge the conventional assumptions of highly specialized, independent attention heads.

- We propose LessIsMore, a training-free sparse attention mechanism featuring: (1) Cross-Head Selection aggregates head-level top-$k$ selections into a unified global ranking, and (2) Stable Recency Window reserves recent contextual information for reasoning coherence.

- We show that LessIsMore matches or improves accuracy on reasoning benchmarks, while speeding up end-to-end decoding by up to $1.6\times$ over full attention. LessIsMore attends to at least $2\times$ fewer tokens than recent sparse attention methods, accelerating attention computation by up to $1.72\times$.

## 2 OBSERVATION

Attention mechanisms are central to the functionality of today's transformer-based LLMs. For each attention head $i$, attention scores and outputs are computed using the scaled-dot product of the query ($Q_i$), key ($K_i$), and value ($V_i$) tensors:

$$W_i = \frac{Q_i K_i^T}{\sqrt{d}}, \quad O_i = \text{softmax}(W_i)V_i \tag{1}$$

Here, $W_i$ represents the attention weights (scores) between tokens, and $O_i$ is the output from the $i$-th attention head.

Sparse attention methods address the computational overhead associated with attending to all tokens by selectively attending to a limited subset. Existing approaches aim to retain tokens most likely to yield high attention scores, constrained by a fixed token budget $k$, through different approximation functions (Tang et al., 2024; Cai et al., 2025; Yang et al., 2024):

$$\arg\max_{\rho} f(Q_i, K_i[\rho], V_i[\rho], k), \quad |\rho| = k \tag{2}$$

where $\rho$ denotes the selected subset of tokens in the KV cache, and the approximation function $f$ is usually an efficient estimation of the underlying ground-truth attention scores to obtain $\rho$. The primary goal of this approximation is to maximize attention recall, defined as the proportion of the ground-truth attention scores that the selected subset covers:

$$R_i = \frac{\sum(\text{softmax}(W_i)[\rho])}{\sum(\text{softmax}(W_i))} \tag{3}$$

A higher attention recall indicates a better coverage of the attention mass with the selected tokens, thereby improving overall accuracy.

### 2.1 LIMITATIONS OF SPARSE ATTENTION IN REASONING

Despite the critical role of attention recall in sparse attention mechanisms, existing methods demonstrate significant limitations when applied to reasoning tasks. Current approaches either misestimate token importance (Tang et al., 2024; Xiao et al., 2023) or focus on locally optimal selections without adequately capturing global attention patterns across layers or decoding steps (Yang et al., 2024).

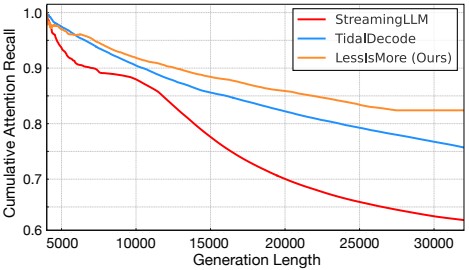 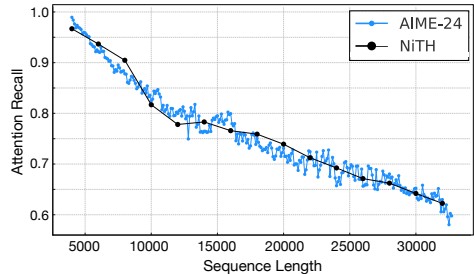

(a) Attention recall of different approaches using 4K token budget on an AIME problem.

(b) Attention recall of retrieval (NiTH) and reasoning (AIME) tasks.

Figure 1: Analysis of attention recall degradation for sparse attention methods on reasoning tasks. Figure 1a compares cumulative average attention recall among StreamingLLM (Xiao et al., 2023), TidalDecode (Yang et al., 2024), and LessIsMore on an AIME-24 reasoning task, using a token budget of 4K and generation length up to 32K tokens on Qwen3-8B. Figure 1b contrasts running-average attention recall of TidalDecode between the reasoning-intensive AIME-24 task throughout the generation and the simpler needle-in-the-haystack retrieval task under the same token budget under various context lengths on Qwen3-8B.

As illustrated in Figure 1a, both StreamingLLM and TidalDecode exhibit substantial attention recall degradation on the AIME-24 task, with degradation becoming more pronounced as decoding length increases. Specifically, TidalDecode reaches only 75% attention recall, while StreamingLLM falls

below 65% due to a more static sparse attention pattern. Additionally, as shown in Figure 1b, although TidalDecode achieves comparable attention recall on both AIME-24 and simpler retrieval tasks (e.g., needle-in-the-haystack), the reasoning tasks inherently involve much longer generation sequences. Consequently, inaccuracies from token selection accumulate over prolonged generation, significantly degrading the reasoning quality in sparse attention. These observations underscore the necessity for designing a selection approach capable of capturing critical tokens globally.

## 2.2 LOCALITIES IN REASONING

Previous studies on traditional tasks have identified the locality property of attention patterns—different decoding layers can share a similar set of critical tokens (Yang et al., 2024). It has also been suggested that different attention heads have distinct functional roles, thereby benefiting from specialized token subsets (Xiao et al., 2024). In Figure 2 below and Figure 8 (Section A.6), we analyze the distribution of the top-4K tokens across all 32 attention heads at different decoding steps using Qwen3-8B, a 36-layer model with GQA. X-axis of Figure 2 is the token's relative position in one decoding step, where the sequence length is 20K and the token position ids will be 0-20K from left to right. The Y-axis shows different query heads, and for each query head's query vector, we obtain the ground truth top-4K tokens by examining its corresponding attention score over all 20K tokens and highlighting them with the highest attention scores. The positional token is simply any token position (any token index along the X-axis) within the 20K past tokens. Selected means it lies in the ground truth top-4K token set, which are highlighted in dark blue. Our analysis extends these findings by demonstrating that attention localities in reasoning tasks manifest both spatially (within the same key-value group and across attention heads) and with recency (across decoding steps).

### 2.2.1 SPATIAL LOCALITY ACROSS ATTENTION HEADS

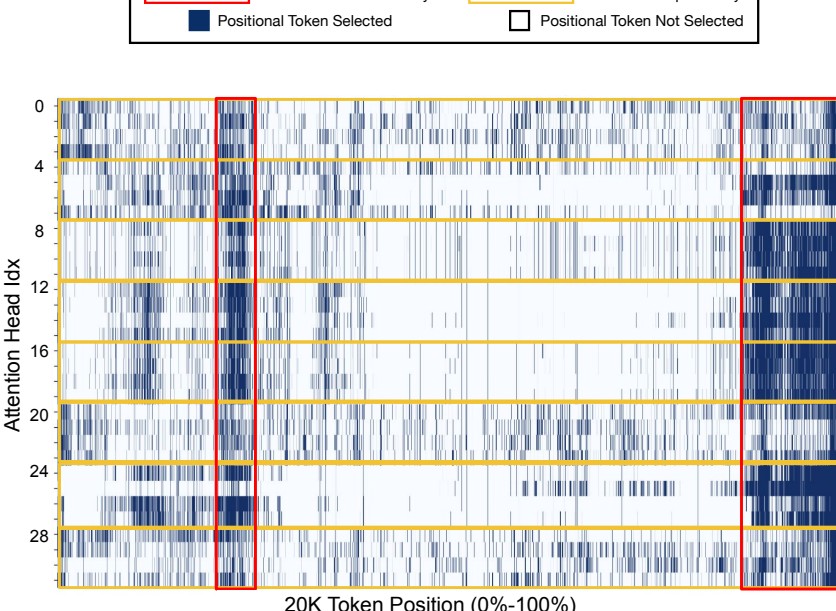

Figure 2: The distribution of the ground-truth top-4K tokens across all attention heads at 20K-th decoding step at Layer 4 on AIME-24 with Qwen3-8B. The dark blue positions stand for tokens included in ground-truth top-4K budget. We enclose the highly overlapped area of attention heads within the same kv group and across all heads with different colors.

The visualization of Figure 2 highlights high overlap among selected tokens across consecutive groups of four attention heads within the same key-value group (yellow regions). Additionally, broader overlaps spanning all attention heads (red regions) include frequently attended recent tokens. This observation contradicts the common belief that each attention head serves specialized functions with

distinct attention score distributions, thereby requiring head-wise token selection with different token subsets for optimal performance. Instead, our findings suggest that reasoning tasks exhibit remarkable consistency in token importance across attention heads, indicating that a cross-head selection strategy may be more effective than maintaining separate token subsets per head in the reasoning process.

### 2.2.2 RECENCY LOCALITY OF RECENT TOKENS

Figure 2 also shows the most recently generated tokens consistently receive high attention scores in subsequent steps. We further conduct comprehensive analysis and observe that the size of this "recency window" remains relatively constant throughout the decoding process (Section A.6).

This recency locality directly reflects the nature of step-by-step reasoning, where each new logical step maintains coherence with immediately preceding conclusions (Lee & Hockenmaier, 2025). Prior work like StreamingLLM (Xiao et al., 2023) has recognized the importance of recent tokens by maintaining a fixed sliding window alongside attention sink tokens. Building on this, our analysis reveals that the ratio between critical recency window size and number of selected tokens remains stable throughout reasoning. This observation supports the design of adaptive token selection mechanisms that allocate a fixed proportional budget to recent tokens to maintain reasoning accuracy efficiently.

## 3 LESSISMORE

---

**Algorithm 1** LessIsMore Decoding Pipeline

---

1: **Input:** Current hidden state $h$, KV cache $\mathcal{C}$, token budget $K$, static ratio $r$
2: **Output:** Logits
3: **Initialize:** $\rho = []$                      ▷ Token buffer for selected indices
4: **for** each decoder layer $i$ **do**
5:      $q, k, v = f(W_{qkv}, h)$
6:      $\mathcal{C}$.append$(k, v)$
7:      **if** $i$ is Full Attention Layer **then**
8:          $o = \text{FullAttention}(q, \mathcal{C}[:])$
9:      **else if** $i$ is Token Selection Layer **then**
10:          $o = \text{FullAttention}(q, \mathcal{C}[:]), P := q \cdot \mathcal{C}.K^\top$     ▷ Full attention and Extract QK product
11:          $\rho_{\text{head}} = \text{TopKIndices}(P[:, : -(K \cdot r)], k = K \cdot (1 - r))$     ▷ Top-k for each head
12:          $\rho_{\text{unified}} = \text{UnionFlatten}(\rho_{\text{head}})$
13:          $\rho_{\text{recent}} = \text{Recent}(K \cdot r)$
14:          $\rho = \rho_{\text{unified}}[: K \cdot (1 - r)] \cup \rho_{\text{recent}}$
15:      **else**
16:          $o = \text{SparseAttention}(q, \mathcal{C}[\rho])$            ▷ Use selected token indices
17:      **end if**
18:      $h = \text{FFN}(o)$
19: **end for**
20: **return** lm_head$(h)$

---

This section introduces LessIsMore, an efficient and effective sparse attention system explicitly designed for reasoning. LessIsMore employs Cross-Head Unified Sparse Attention (CUSA) by leveraging the locality attention patterns identified in Section 2.2.1 and Section 2.2.2 by integrating two key techniques: Cross-Head Selection and Stable Recency Window. In this paper, LessIsMore adopts TidalDecode (Yang et al., 2024), one of the best-performing sparse attention methods, as a backbone; specifically, it starts with two full attention layers, includes two dedicated token selection layers, and employs sparse attention in the remaining layers. However, as discussed in Section 1 and shown in Section 4.4.1, LessIsMore's underlying techniques can be effectively incorporated into any sparse attention using approximation algorithms. The detailed mechanism of token selection and sparse attention layers in LessIsMore is formalized in Algorithm 1.

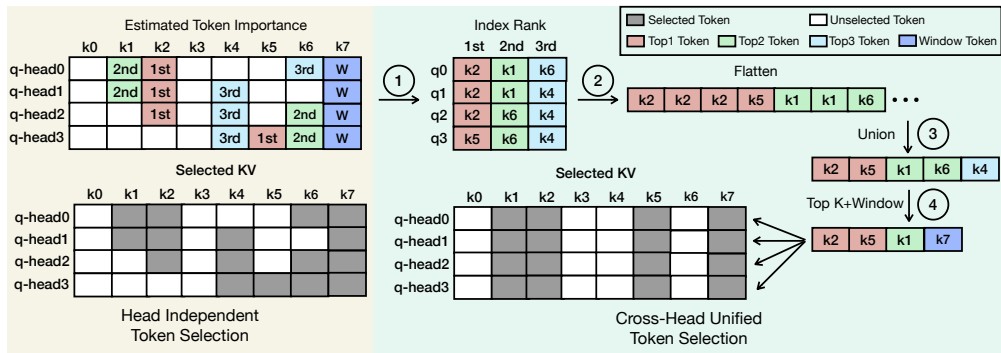

Figure 3: Overview of CUSA in LessIsMore. Each attention head first selects its top-$k$ tokens under budget $K = 4$ with $r = 0.25$ reserved for the recency window (Stable Recency Window), producing head-level selections $\rho_{head}$. These indices are then aggregated across heads through Cross-Head Selection into a unified ranked set, from which the top entries are kept. Finally, this unified set is concatenated with the most recent tokens to form the final token set $\rho$, which is shared by all sparse attention layers. Only tokens in $\rho$ are loaded from the KV cache until the next selection step, ensuring both cross-head consistency and stable recency preservation.

## 3.1 CROSS-HEAD UNIFIED SPARSE ATTENTION (CUSA)

As shown in lines 4–19 of Algorithm 1, LessIsMore processes each decoding step using three distinct layer types. In full attention layers (lines 7–8), standard attention computation is performed. In token selection layers (lines 9–15), as illustrated in Figure 3, LessIsMore selects tokens and performs attention computation through CUSA. The overall token budget $K$ is divided into two subsets: top-$k$ tokens and the most recent tokens. During token selection, full attention is first applied to compute the QK-product $P = q \cdot CK^T$ (line 10), which guides the token importance estimation.

LessIsMore then applies Cross-Head Selection through the unified token selection process (lines 11-14), excluding the most recent tokens to focus the selection process on historical context. Each attention head independently selects its top-k token indices based on attention scores (line 11), followed by global aggregation and sorting across all heads (line 12). The top-ranked indices are then combined with the recent token indices determined by Stable Recency Window (lines 13-14) to form the final selected token set $\rho$. LessIsMore then fetches the corresponding unified key-value tensors from the KV cache for the selected indices. The indices are shared by all subsequent layers for CUSA (line 16) until the next selection layer or the end of the current decoding step.

### 3.1.1 CROSS-HEAD SELECTION

Cross-Head Selection aims to take advantage of the spatial locality observed with the token attention of Section 2.2.1 to improve efficiency and effectiveness. The core mechanism is implemented following lines 11-12 of Algorithm 1, where each attention head independently computes attention scores and identifies the top-k tokens most relevant to its query through TopKIndices$(P[:, : -(K \cdot r)], k = K \cdot (1 - r))$. Instead of maintaining separate sets of tokens per head, which increases the selection overhead and complexity of KV cache access, Cross-Head Selection aggregates the independently selected tokens from all attention heads.

The aggregation process, formalized as UnionFlatten$(\rho_{head})$ in line 12, flattens the top-k token indices selected by each head into a single unified set. This combined set is then globally sorted according to the tokens' rank within its attention head. Subsequently, only the globally highest-ranked tokens, limited by the predefined token budget $K \cdot (1 - r)$ (line 14), are retained. This unified selection strategy not only improves the attention recall shown in Figure 7 by using the observed spatial location, where tokens frequently overlap in importance across heads, but also significantly simplifies token retrieval, enhancing computational efficiency during sparse attention computation.

### 3.1.2 STABLE RECENCY WINDOW

Stable Recency Window addresses the consistent recency locality observed in reasoning tasks, where recently generated tokens are repeatedly and consistently attended by all attention heads in Figure 8. To exploit this pattern, Stable Recency Window dedicates a fixed ratio of the total token budget $K$ exclusively to the most recently generated tokens, forming a "stable recency window." This mechanism is implemented in lines 13-14 of Algorithm 1.

Prior sparse attention training approaches maintain a constant number of tokens as the sliding window size regardless of token budgets (Yuan et al., 2025). In contrast, the stable recency window in LessIsMore is determined by a predefined ratio $r$, typically a small fraction of $K$, through Recent($K \cdot r$) (line 13). The final set of unified cross-head tokens used for sparse attention computation is formed by the union operation $\rho = \rho_{unified}[: K \cdot (1 - r)] \cup \rho_{recent}$ (line 14), consisting of the selected cross-head top-k tokens and the most recent tokens in the window. This design directly reflects the empirical observation that recently generated tokens possess critical contextual information necessary for accurate and coherent step-by-step reasoning. By explicitly allocating resources to recent tokens via this algorithmic approach, Stable Recency Window effectively ensures high attention recall and improved reasoning quality while maintaining computational efficiency (Section 4.4.2).

## 4 EXPERIMENTS

### 4.1 EXPERIMENT SETUP

We conduct extensive experiments to evaluate the accuracy and efficiency of LessIsMore. Our experiments consider four widely-used reasoning models from two families, Qwen3-4, Qwen3-8B, and Qwen3-14B (Team, 2025) and DeepSeek-R1-Distill-Llama-8B (DeepSeek-AI, 2025) backed up with GQA. All models are specifically trained for reasoning tasks and perform the effective thinking process by generating extensive tokens. Further, we evaluate on multiple mainstream reasoning tasks, including AIME-24, AIME-25, GPQA-Diamond, and MATH500.

In Section 4.2, we compare LessIsMore with both training-free (TidalDecode, Quest) and training-required (SeerAttention-r) sparse attention baselines. For training-free approaches, we keep the first two full attention layers for TidalDecode. To ensure a fair comparison, we set the same selection layer for LessIsMore and TidalDecode - Layer 12 for DeepSeek-R1-8B, Qwen3-8B, and Qwen3-14B; Layer 20 for Qwen3-4B.[1] The static ratio $r$ of LessIsMore is set to $0.25$ and 4 tokens are always reserved for attention sink in this section. For Quest, we maintain the hybrid attention layers and block size of 16 on DeepSeek-R1-8B with apply all sparse attention layers and block size of 32 on Qwen model family. For SeerAttention-r, which only releases Qwen models, we maintain the same experiment configuration of in (Gao et al., 2025), where the block size is set to 64 and all layers perform sparse attention. To guarantee the consistency of evaluation results, we evaluate all models with a maximum generation length of 32K and the same instruction prompt on each reasoning tasks. For all experiments, we generate, 64, 8, and 16 answers for each problem in AIME-24/25, MATH500, GPQA-Diamond datasets, respectively, and compute the Pass@1 accuracy.

In Section 4.3, we compare decoding efficiency for LessIsMore implemented with customized kernels for the selection-based model (TidalDecode) and full attention with FlashInfer (Ye et al., 2024).

### 4.2 EVALUATION ON REASONING TASKS

Figure 4 presents the accuracy comparison among Full Attention, LessIsMore, and other sparse attention methods as baselines across mainstream reasoning benchmarks, AIME-24, AIME-25, MATH500, and GPQA-Diamond. The first three are from the complex mathematical contests, and GPQA-Diamond compromises of graduate-level STEM proofing problems. All datasets are evaluated on the reasoning-focused language models DeepSeek-R1-8B and Qwen3-4B, -8B, and -14B. For challenging AIME-24/25 tasks, experiments span token budgets of 1K, 2K, 4K, 6K, and 8K where the Full Attention model solves problems with an average reasoning length of 15K and 17K tokens, respectively. In contrast, existing sparse attention methods not only suffer accuracy degradation but also extend generation lengths significantly, often requiring 15K–30K tokens to

---

[1]The choice is justified in Section A.4. Table 11 summarizes re-selection-layer choice for each model.

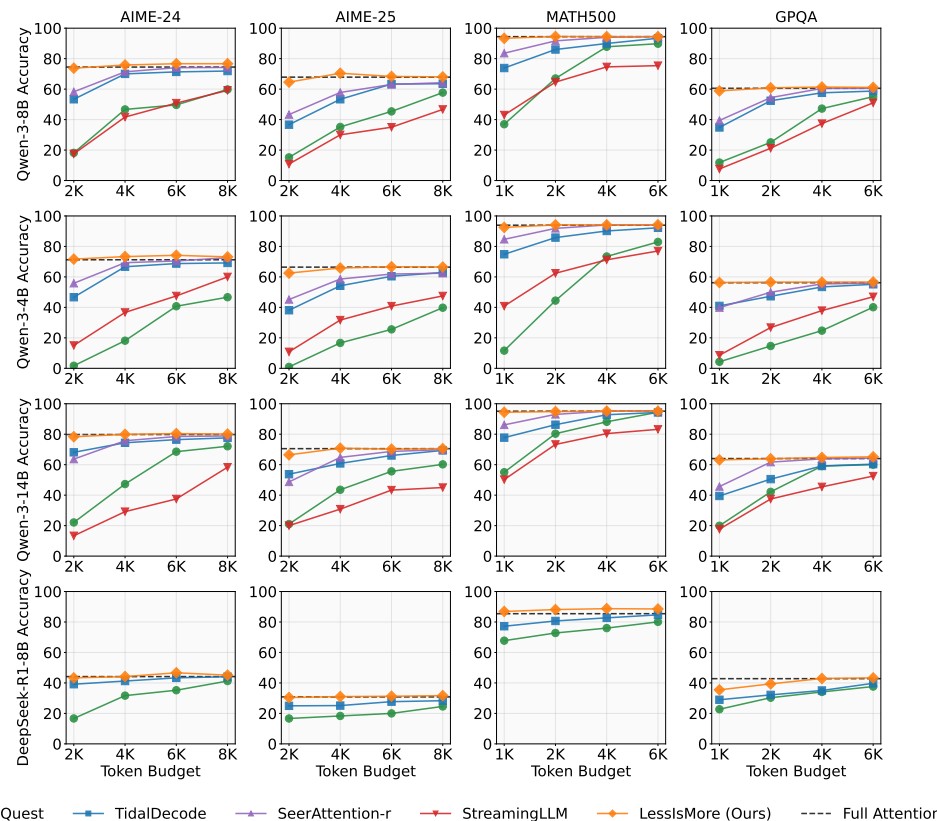

Figure 4: Accuracy results of LessIsMore (ours), Quest, StreamingLLM, TidalDecode, SeerAttention-r, and Full Attention across for multiple main-stream reasoning tasks. LessIsMore consistently achieves the lossless accuracy with small token budgets (1K or 2K), always outperforming all others.

complete the same problems. A similar trend holds for GPQA (Full Attention 8K vs. 8K–18K under baselines) and MATH500 (Full-Attention 5K vs. 5K–16K under baselines). By comparison, LessIsMore consistently achieves the highest accuracy across all evaluated tasks and token budgets, while maintaining generation lengths nearly identical to Full Attention (15K for AIME, 8K for GPQA, 5K for MATH). Specifically, for Qwen3-8B on AIME-24 at the smallest budget (2K tokens), LessIsMore attains nearly lossless accuracy, surpassing Quest, TidalDecode, and training-required SeerAttention-r, all of which suffer notable degradation and longer reasoning traces. This dual advantage—accuracy preservation without length inflation—underscores LessIsMore 's ability to retain critical contextual information and facilitate fast, accurate reasoning with limited token budgets.

## 4.3 EFFICIENCY EVALUATION

To evaluate the practical efficiency gains of LessIsMore, we implement customized kernels on top of the state-of-the-art attention kernel library (Ye et al., 2024)for GQA-based models. We conduct both end-to-end time-between-token (TBT) and kernel-level latency analysis under practical serving setups and report the results in Figure 5. Our evaluation uses DeepSeek-R1-Distill-LLama-8B (AI, 2024b; DeepSeek-AI, 2025) deployed on a single 80GB NVIDIA A100 GPU.

For end-to-end speed-up gains, as shown in Figure 5a, compared to the Full Attention baseline, LessIsMore consistently accelerates decoding, delivering **1.1×**, **1.3×**, and **1.6×** end-to-end speedups at 16K, 32K, and 64K context lengths, respectively. These improvements highlight LessIsMore 's ability to preserve reasoning quality while providing meaningful computational savings. Given that modern reasoning models—both open- and closed-source—already support generation lengths well beyond 100K tokens (OpenAI, 2025b; Anthropic, 2025; OpenAI, 2025a), the efficiency advantages of LessIsMore are expected to scale even further in real-world deployments. For kernel-level speed-up

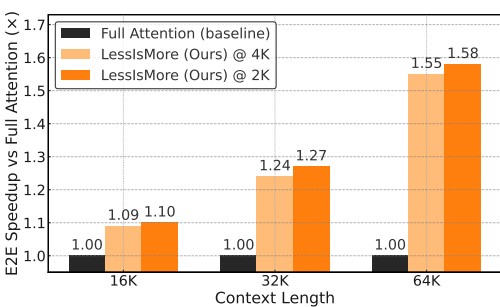
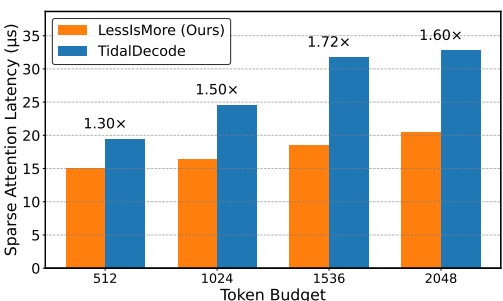

(a) End-to-end Time-Between-Token (TBT) speedup under different context lengths and token budgets. **Higher speed-up is better**. LessIsMore can achieve speed-up ranging from $1.09\times$-$1.58\times$ compared with the full attention baseline due to attention sparsity.

(b) Sparse attention kernel latency comparison between LessIsMore and TidalDecode under different token budgets. **Lower latency is better**, and LessisMore can achieve a speed-up ranging from $1.3\times$-$1.72\times$ consistently across all token budgets.

Figure 5: Efficiency results with DeepSeek-R1-Distill-LLama-8B on one 80GB NVIDIA A100 GPU.

gains, as shown in Figure 5b, we observe that LessIsMore can consistently achieve speed-ups from $\mathbf{1.3\times}$ to $\mathbf{1.7\times}$ on the sparse attention computation, compared to the TidalDecode approach under the same token budgets. This is mainly due to LessIsMore's unified token selection design, which is more kernel-friendly for GQA-based models. More specifically, methods like TidalDecode/Quest require more KV loading per KV group when different query heads can select a different set of tokens.

## 4.4 ABLATION STUDY[2]

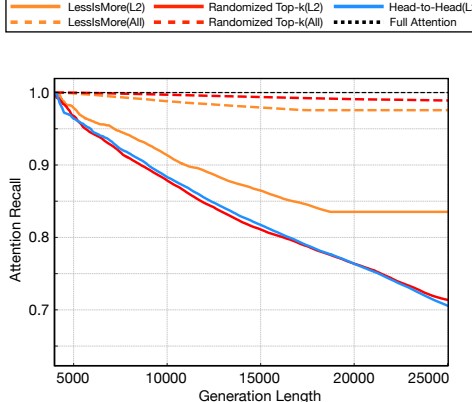
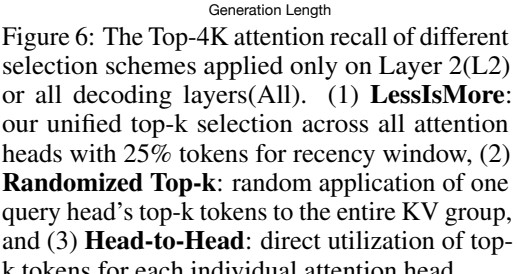

Figure 6: The Top-4K attention recall of different selection schemes applied only on Layer 2(L2) or all decoding layers(All). (1) **LessIsMore**: our unified top-k selection across all attention heads with 25% tokens for recency window, (2) **Randomized Top-k**: random application of one query head's top-k tokens to the entire KV group, and (3) **Head-to-Head**: direct utilization of top-k tokens for each individual attention head

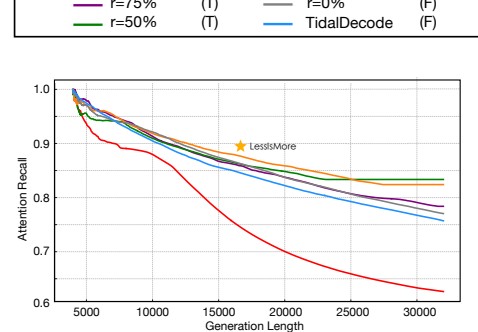

Figure 7: Ablation study on the impact of varying static recency window ratio $r$ in LessIsMore ($\star$) on the AIME-24 reasoning task, using a token budget of 4K and generation length up to 32K tokens on Qwen3-8B. LessIsMore corresponds to the 25% recent setting combined with Cross-Head Selection, (labeled with ($\star$)). We compare it against alternative recent window ratios, the 100% recent baseline (i.e., using only recent tokens), and TidalDecode.

### 4.4.1 EFFECTIVENESS OF LESSISMORE'S AGGREGATION ON GQA

To assess whether LessIsMore's unified selection generalizes to GQA models, we compare three aggregation strategies on Qwen3-8B with shared KV heads: **LessIsMore**, **Randomized Top-k**,

---

[2]Moved from Appendix during rebuttal period, originally in Appendix Section A.5.1 and Section A.5.3.

and **Head-to-Head** (Figure 6). When selection is applied at all decoding layers, locally optimized schemes such as Randomized Top-k appear competitive. However, when selection is reduced to only Layer 2 (a more realistic low-frequency setting), these local heuristics fail to generalize, leading to substantially lower attention recall. In contrast, LessIsMore maintains strong recall in both settings, demonstrating that a globally consistent cross-head selection strategy is significantly more robust than layer-specific or head-specific methods. This underscores the importance of unified aggregation for stable token importance estimation under sparse selection.

### 4.4.2 EFFECT OF RECENT WINDOW RATIO

We analyze how the recent-window ratio $r$ affects attention recall and correctness on AIME-24 under a 4K token budget (Figure 7). Only configurations that combine a recent window with Cross-Head Selection (25%, 50%, 75%) successfully solve the task. Using only recent tokens ($r = 100\%$) yields the lowest recall by discarding essential long-range context. TidalDecode improves recall yet still fails to reach the correct answer, and using Cross-Head Selection with 0% recent further improves recall but likewise fails. Introducing even a modest recent window consistently boosts recall. The 25% configuration—corresponding to the LessIsMore design—achieves the highest recall across the generation, validating the choice of allocating a small fraction of the token budget to recent tokens.

## 5 RELATED WORK

**Efficient sparse attention with KV cache compression.** Sparse attention mechanisms reduce the computational overhead and memory requirements of attention calculations by selectively attending to only a subset of tokens, significantly improving inference efficiency for long-sequence tasks (Yang et al., 2024; Tang et al., 2024). Current approaches can be broadly categorized into two main paradigms: eviction-based and selection-based methods. Eviction-based approaches (Xiao et al., 2023; Zhang et al., 2023; Li et al., 2024; Adnan et al., 2024) permanently discard tokens from the KV cache based on predefined criteria, achieving better memory savings by maintaining a smaller cache size throughout generation. In contrast, selection-based methods (Yang et al., 2024; Tang et al., 2024; Hao et al., 2025; Liu et al., 2024) retain the full KV cache but dynamically choose which tokens to attend to during computation, typically optimizing for locally maximal attention scores and choosing different tokens for each attention head. While both approaches are effective on standard long-context tasks such as retrieval and summarization, they face significant challenges when applied to reasoning tasks due to the accumulation of selection errors over extended generation sequences.

**Sparse attention in reasoning.** Recent reasoning models leverage the principle that scaling test-time compute can be more effective than scaling model parameters (Wei et al., 2023; DeepSeek-AI, 2025), generating extensive token sequences to enhance reasoning accuracy through deliberative processes. However, the lengthened generation nature of reasoning tasks poses unique challenges for applying existing sparse attention methods. When applied on reasoning tasks, prior works either suffer from significant accuracy degradation when using small token retention ratios (Yang et al., 2024; Tang et al., 2024; Cai et al., 2025) or require computationally expensive post-training procedures to mitigate accuracy loss accumulated during generation (Gao et al., 2025), both of which also significantly increase the generation length of reasoning tasks. In contrast, LessIsMore is proposed as a selection-based, training-free approach that leverages intrinsic spatial and recency attention patterns within the reasoning process to achieve high accuracy with substantially reduced token utilization ratios but without extending the generation length.

## 6 CONCLUSION

We introduced LessIsMore, a training-free sparse attention mechanism tailored for reasoning tasks. With Cross-Head Unified Sparse Attention, LessIsMore addresses the accuracy and length inflation common in prior methods with efficiency win. Experiments show that it maintains near-lossless accuracy at high sparsity-e.g., full accuracy on AIME-24 with only a 2K budget-while reducing latency. With optimized customized kernels, LessIsMore achieves up to $1.6\times$ end-to-end decoding speed-ups over full attention and accelerates sparse attention computation by up to $1.72\times$ compared to prior state-of-the-art methods. These results highlight the effectiveness of exploiting global attention patterns for faster, more accurate reasoning.

REPRODUCIBILITY STATEMENT

The code and scripts necessary to fully reproduce both the accuracy and efficiency experiments presented in this paper are included in the supplementary materials. Detailed experiment settings and hyperparameters are described in Section 4 and Section A, while additional ablation studies are provided in Section A.5. Together, these resources enable faithful reproduction of all reported findings.

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

# A APPENDIX

## A.1 REASONING EVALUATION

### A.1.1 REASONING EVALUATION RESULTS ON QWEN3-4B, QWEN3-8B, QWEN3-14B, AND DEEPSEEK-R1-DISTILL-LLAMA-8B

In Table 1 and Table 2, we record pass@1 accuracy plotted in Figure 4 with more sparse attention baselines and token budgets.

### A.1.2 REASONING EVALUATION RESULTS ON QWEN3-4B, QWEN3-8B, AND QWEN3-14B WITH LARGE SAMPLE SIZE

In Table 3 and Table 4, we further evaluate pass@1 accuracy of LessIsMore and TidalDecode with 64 sample per query for AIME24/25 and 16 samples for GPQA-Diamond, reporting the difference compared to accuracy in Table 1 and Table 2 accordingly. With larger sample sizes, LessIsMore still always achieves the highest accuracy among all evaluated sparse attention approaches and stays closely to Full Attention baseline across all token budgets, models, and reasoning tasks.

Moreover, to demonstrate the variance in reasoning evaluation, we generate a pool of 512, 64, and 32 answers per problem in AIME-24/25, GPQA, and MATH500 with Full Attention on Qwen3-8B, respectively. We calculate the variance of different sample sizes in Table 5 by sampling outputs from the pool and computing the variance of final accuracy over 100 runs, where it shows the evaluated AIME-24/25, GPQA, and MATH500 exhibit minimal variance when evaluated on sample size of 64, 16, and 8, respectively.

## A.2 LONG-CONTEXT EVALUATION

### A.2.1 NEEDLE-IN-THE-HAYSTACK

Table 6 shows that on both non-reasoning models Llama-3-8B-Instruct-Gradient-1048k (AI, 2024a) and Llama-3.1-8B-Instruct (AI, 2024b), LessIsMore maintains strong long-context retrieval performance, consistently outperforming Quest and matching or exceeding TidalDecode even at very small token budgets. LessIsMore and TidalDecode both apply Layer 13 as re-selection layer. Remarkably, LessIsMore achieves full Needle-in-the-Haystack accuracy with only 32–128 tokens with up to 100K context (0.1–0.3% of the input), demonstrating that LessIsMore remains effective for long-context retrieval tasks and generalizes well beyond reasoning-oriented models.

### A.2.2 EVALUATION OF LONGBENCH

Evaluation of LessIsMore, Quest, and TidalDecode on LongBench datasets (Bai et al., 2023) is shown in Table 7. As a result, LessIsMore consistently achieves higher average F1 across five LongBench datasets MultiFieldQA, Qasper, HotpotQA, TriviaQA, PassageRetrieval, matching or surpassing the Full Attention baseline while using only a 4K token budget and even achieve the highest average score. These results highlight that LessIsMore not only preserves accuracy in complicated reasoning tasks but also exhibits potential on solving long-context tasks.

## A.3 EFFICIENCY EVALUATION

### A.3.1 END-TO-END SPEEDUP ON INFERENCE ENGINE

To evaluate the practical impact of LessIsMore under modern serving infrastructures, we additionally measure end-to-end decoding performance when integrating our method into the SGLang serving stack (Zheng et al., 2024), which is built on top of the FlashInfer attention kernel library. Since FlashInfer is also used by vLLM (Kwon et al., 2023) and provides state-of-the-art fused attention kernels, this experiment reflects realistic deployment conditions. All baseline methods, including TidalDecode, are also implemented using FlashInfer kernels to ensure a fair comparison.

Table 8 reports the time-between-token (TBT) speed-up achieved when applying LessIsMore with different token budgets under 16K, 32K, and 64K context lengths. Across all settings, LessIsMore consistently accelerates end-to-end decoding, reaching up to $1.51\times$ speed-up at 64K context length.

Table 1: Results of 2K-, 4K-, 6K-, 8K-, and 16K-token-budget in Figure 4 evaluated on Qwen3-4B, Qwen3-8B, Qwen3-14B, and DeepSeek-R1-Distill-Llama-8B for AIME24 and AIME25 benchmarks. The highest accuracy of all sparse attention approaches for each token budget and approach are in bold.

| Model (Task) | Method / Budget | K=2000 | K=4000 | K=6000 | K=8000 |
|---|---|---|---|---|---|
| Qwen-3-4B (AIME-24) | Quest | 1.67 | 18.14 | 40.74 | 46.67 |
| | TidalDecode | 46.67 | 66.67 | 68.75 | 69.17 |
| | SeerAttention-r | 55.83 | 69.32 | 70.47 | 72.60 |
| | StreamingLLM | 15.00 | 36.67 | 47.50 | 60.00 |
| | **LessIsMore (Ours)** | **71.67** | **73.33** | **74.17** | **73.12** |
| | Full Attention | 71.25 | 71.25 | 71.25 | 71.25 |
| Qwen-3-4B (AIME-25) | Quest | 0.92 | 16.67 | 25.56 | 39.79 |
| | TidalDecode | 38.12 | 54.17 | 60.41 | 62.92 |
| | SeerAttention-r | 45.16 | 58.59 | 61.88 | 62.40 |
| | StreamingLLM | 10.83 | 31.67 | 40.83 | 47.50 |
| | **LessIsMore (Ours)** | **62.50** | **65.83** | **66.67** | **66.46** |
| | Full Attention | 66.41 | 66.41 | 66.41 | 66.41 |
| Qwen-3-8B (AIME-24) | Quest | 18.15 | 46.67 | 49.63 | 59.79 |
| | TidalDecode | 53.33 | 70.00 | 71.30 | 71.87 |
| | SeerAttention-r | 58.23 | 71.35 | 74.06 | 74.22 |
| | StreamingLLM | 17.5 | 41.67 | 50.83 | 59.17 |
| | **LessIsMore (Ours)** | **73.75** | **75.83** | **76.67** | **76.67** |
| | Full Attention | 74.48 | 74.48 | 74.48 | 74.48 |
| Qwen-3-8B (AIME-25) | Quest | 15.2 | 35.18 | 45.37 | 57.71 |
| | TidalDecode | 36.67 | 53.33 | 63.33 | 63.54 |
| | SeerAttention-r | 43.30 | 57.81 | 63.07 | 64.22 |
| | StreamingLLM | 10.83 | 30.00 | 35.00 | 46.67 |
| | **LessIsMore (Ours)** | **64.58** | **70.42** | **68.33** | **68.02** |
| | Full Attention | 67.86 | 67.86 | 67.86 | 67.86 |
| Qwen-3-14B (AIME-24) | Quest | 22.08 | 47.29 | 68.54 | 72.08 |
| | TidalDecode | 68.12 | 74.38 | 76.45 | 77.60 |
| | SeerAttention-r | 63.65 | 75.73 | 78.49 | 78.85 |
| | StreamingLLM | 13.33 | 29.17 | 37.50 | 58.33 |
| | **LessIsMore (Ours)** | **78.33** | **79.90** | **80.41** | **80.10** |
| | Full Attention | 79.79 | 79.79 | 79.79 | 79.79 |
| Qwen-3-14B (AIME-25) | Quest | 21.04 | 43.54 | 55.62 | 60.21 |
| | TidalDecode | 53.75 | 60.83 | 66.04 | 69.37 |
| | SeerAttention-r | 48.70 | 64.79 | 68.70 | 69.84 |
| | StreamingLLM | 20.00 | 30.83 | 43.33 | 45.00 |
| | **LessIsMore (Ours)** | **66.45** | **70.83** | **70.21** | **70.52** |
| | Full Attention | 70.52 | 70.52 | 70.52 | 70.52 |
| DeepSeek-R1-Distill-Llama-8B (AIME-24) | Quest | 16.67 | 31.67 | 35.21 | 41.25 |
| | TidalDecode | 39.16 | 41.25 | 43.33 | 44.11 |
| | **LessIsMore (Ours)** | **43.33** | **44.16** | **46.67** | **45.10** |
| | Full Attention | 44.16 | 44.16 | 44.16 | 44.16 |
| DeepSeek-R1-Distill-Llama-8B (AIME-25) | Quest | 16.67 | 18.33 | 20.00 | 24.58 |
| | TidalDecode | 24.97 | 25.14 | 27.71 | 28.33 |
| | **LessIsMore (Ours)** | **30.42** | **31.04** | **31.25** | **31.67** |
| | Full Attention | 30.83 | 30.83 | 30.83 | 30.83 |

### A.3.2 END-TO-END SPEED-UP VS. SPARSE ATTENTION BASELINES

To complement the kernel-level analysis in the main paper, we additionally report end-to-end decoding latency when integrating LessIsMore and other sparse attention baselines into a modern serving stack.

Table 2: Results of 1K-, 2K-, 4K-, 6K-, and 8K-token-budget in Figure 4 evaluated on Qwen3-4B, Qwen3-8B, Qwen3-14B, and DeepSeek-R1-Distill-Llama-8B for MATH500 and GPQA benchmarks. The highest accuracy of all sparse attention approaches for each token budget and approach are in bold.

| Model (Task) | Method / Budget | K=1000 | K=2000 | K=4000 | K=6000 |
|---|---|---|---|---|---|
| Qwen-3-4B (MATH500) | Quest | 11.62 | 44.45 | 73.40 | 82.95 |
| | TidalDecode | 74.85 | 85.80 | 90.15 | 92.22 |
| | SeerAttention-r | 84.67 | 91.85 | 94.10 | 94.12 |
| | StreamingLLM | 40.84 | 62.40 | 71.20 | 77.00 |
| | **LessIsMore (Ours)** | **92.50** | **94.12** | **94.16** | **94.22** |
| | Full Attention | 93.93 | 93.93 | 93.93 | 93.93 |
| Qwen-3-4B (GPQA) | Quest | 4.29 | 14.64 | 24.74 | 40.10 |
| | TidalDecode | 40.97 | 47.28 | 53.41 | 55.14 |
| | SeerAttention-r | 39.84 | 49.94 | 55.40 | 55.90 |
| | StreamingLLM | 8.58 | 26.76 | 37.87 | 46.96 |
| | **LessIsMore (Ours)** | **56.31** | **56.56** | **56.56** | **56.64** |
| | Full Attention | 56.19 | 56.19 | 56.19 | 56.19 |
| Qwen-3-8B (MATH500) | Quest | 36.95 | 66.98 | 87.80 | 89.80 |
| | TidalDecode | 73.85 | 86.00 | 89.95 | 93.38 |
| | SeerAttention-r | 83.57 | 91.67 | 94.00 | 94.53 |
| | StreamingLLM | 43.00 | 64.60 | 74.60 | 75.40 |
| | **LessIsMore (Ours)** | **93.35** | **94.55** | **94.45** | **94.42** |
| | Full Attention | 94.43 | 94.43 | 94.43 | 94.43 |
| Qwen-3-8B (GPQA) | Quest | 11.80 | 25.06 | 47.22 | 55.05 |
| | TidalDecode | 34.84 | 52.39 | 57.57 | 58.71 |
| | SeerAttention-r | 39.43 | 54.41 | 60.48 | 60.57 |
| | StreamingLLM | 7.58 | 21.21 | 37.37 | 51.01 |
| | **LessIsMore (Ours)** | **58.84** | **60.86** | **61.36** | **61.11** |
| | Full Attention | 60.54 | 60.54 | 60.54 | 60.54 |
| Qwen-3-14B (MATH500) | Quest | 55.05 | 80.35 | 88.15 | 94.40 |
| | TidalDecode | 77.75 | 86.25 | 92.78 | 94.28 |
| | SeerAttention-r | 86.12 | 93.02 | 95.12 | 95.40 |
| | StreamingLLM | 50.21 | 73.28 | 80.40 | 83.20 |
| | **LessIsMore (Ours)** | **94.40** | **94.85** | **95.14** | **95.12** |
| | Full Attention | 95.10 | 95.10 | 95.10 | 95.10 |
| Qwen-3-14B (GPQA) | Quest | 19.92 | 42.17 | 59.10 | 60.02 |
| | TidalDecode | 39.47 | 50.53 | 59.15 | 60.35 |
| | SeerAttention-r | 45.64 | 61.68 | 63.83 | 64.33 |
| | StreamingLLM | 17.70 | 37.47 | 45.45 | 52.52 |
| | **LessIsMore (Ours)** | **63.13** | **63.89** | **64.71** | **65.15** |
| | Full Attention | 64.02 | 64.02 | 64.02 | 64.02 |
| DeepSeek-R1-Distill-Llama-8B (MATH500) | Quest | 67.75 | 72.78 | 75.99 | 80.08 |
| | TidalDecode | 77.21 | 80.68 | 82.72 | 84.68 |
| | **LessIsMore (Ours)** | **86.90** | **88.15** | **88.75** | **88.54** |
| | Full Attention | 85.45 | 85.45 | 85.45 | 85.45 |
| DeepSeek-R1-Distill-Llama-8B (GPQA) | Quest | 22.73 | 30.30 | 34.09 | 37.63 |
| | TidalDecode | 28.96 | 32.16 | 35.04 | 39.77 |
| | **LessIsMore (Ours)** | **35.47** | **39.39** | **42.92** | **43.31** |
| | Full Attention | 42.80 | 42.80 | 42.80 | 42.80 |

All methods were evaluated using the DeepSeek-R1-Distilled-LLaMA-8B model on a single NVIDIA A5000 GPU. For each method, we measure the per-token decoding latency under a token budget of 2K and context lengths of 16K, 32K, and 64K tokens. The backend attention computation is executed

Table 3: Results of 2K-, 4K-, 6K-, and 8K-token-budget with sampling 64 answers by problem evaluated on Qwen3-4B, Qwen3-8B, and Qwen3-14B, and DeepSeek-R1-Distill-Llama-8B for AIME24 and AIME25 benchmarks. For TidalDecode and LessIsMore, we show 64-sample pass@1 as new accuracy ($\Delta$ vs. 16-sample in Table 1). Full Attention is shown for reference.

| Model (Task) | Method / Budget | K=2000 | K=4000 | K=6000 | K=8000 |
|---|---|---|---|---|---|
| Qwen-3-4B (AIME-24) | TidalDecode | 46.89 (+0.22) | 67.13 (+0.46) | 68.32 (-0.43) | 69.17 |
| | **LessIsMore (Ours)** | **71.48 (-0.19)** | **73.03 (-0.30)** | **74.37 (+0.20)** | **73.12** |
| | Full Attention | 71.25 | 71.25 | 71.25 | 71.25 |
| Qwen-3-4B (AIME-25) | TidalDecode | 38.77 (+0.65) | 53.17 (-1.00) | 60.57 (+0.16) | 62.92 |
| | **LessIsMore (Ours)** | **62.87 (+0.37)** | **65.94 (+0.11)** | **66.56 (-0.11)** | **66.46** |
| | Full Attention | 66.41 | 66.41 | 66.41 | 66.41 |
| Qwen-3-8B (AIME-24) | TidalDecode | 53.18 (-0.15) | 69.90 (-0.10) | 71.57 (+0.27) | 71.87 |
| | **LessIsMore (Ours)** | **73.00 (-0.75)** | **75.56 (-0.27)** | **76.45 (-0.22)** | **76.67** |
| | Full Attention | 74.48 | 74.48 | 74.48 | 74.48 |
| Qwen-3-8B (AIME-25) | TidalDecode | 37.31 (+0.64) | 53.82 (+0.49) | 62.70 (-0.63) | 63.54 |
| | **LessIsMore (Ours)** | **65.24 (+0.66)** | **70.31 (-0.11)** | **68.00 (-0.33)** | **68.02** |
| | Full Attention | 67.86 | 67.86 | 67.86 | 67.86 |
| Qwen-3-14B (AIME-24) | TidalDecode | 67.85 (-0.27) | 75.25 (+0.87) | 76.14 (-0.31) | 77.60 |
| | **LessIsMore (Ours)** | **78.58 (+0.25)** | **80.39 (+0.49)** | **80.19 (-0.22)** | **80.10** |
| | Full Attention | 79.79 | 79.79 | 79.79 | 79.79 |
| Qwen-3-14B (AIME-25) | TidalDecode | 53.33 (-0.42) | 59.83 (-1.00) | 65.59 (-0.45) | 69.37 |
| | **LessIsMore (Ours)** | **66.56 (+0.11)** | **70.59 (-0.24)** | **70.48 (+0.27)** | **70.52** |
| | Full Attention | 70.52 | 70.52 | 70.52 | 70.52 |
| Llama-8B (AIME-24) | TidalDecode | 39.41 (+0.25) | 41.82 (+0.57) | 43.89 (+0.56) | 44.11 |
| | **LessIsMore (Ours)** | **43.22 (-0.11)** | **44.28 (+0.12)** | **45.84 (-0.83)** | **45.10** |
| | Full Attention | 44.16 | 44.16 | 44.16 | 44.16 |
| Llama-8B (AIME-25) | TidalDecode | 25.31 (+0.34) | 25.77 (+0.63) | 27.85 (+0.14) | 28.33 |
| | **LessIsMore (Ours)** | **29.93 (-0.49)** | **31.91 (+0.87)** | **31.93 (+0.68)** | **31.67** |
| | Full Attention | 30.83 | 30.83 | 30.83 | 30.83 |

through SGLang with FlashInfer kernels, providing a unified and optimized execution environment for all baselines.

Table 9 summarizes the results. Across all context lengths, LessIsMore achieves consistently lower latency than TidalDecode and Quest. This improvement stems from the kernel-friendly design of LessIsMore, particularly its unified cross-head aggregation, which reduces memory movement and KV loading overhead in GQA-based models. For SeerAttention-R, no public end-to-end implementation is currently available; however, given its block-level sparsity pattern, its latency is expected to be comparable to Quest.

### A.3.3 KERNEL-LEVEL FLOP, MEMORY TRANSFER, AND LATENCY ANALYSIS

To complement the end-to-end evaluations, we report kernel-level metrics that highlight the efficiency benefits of LessIsMore relative to other sparse attention methods. Using the DeepSeek-R1-Distilled-LLaMA-8B model with a token budget of 2K and a context length of 16K, we profile the FLOPs, global-to-shared memory data transfer, on-device memory consumption, and per-kernel latency for the attention computation. All measurements are obtained using FlashInfer as the backend attention kernel library. Table 10 shows that even if the FLOP count is identical across sparse attention methods, LessIsMore performs significantly less global-to-shared memory transfer than TidalDecode or Quest/SeerAttention-R due to its unified cross-head token selection, which reduces redundant KV loading across attention heads. This reduction directly contributes to our lower kernel latency. StreamingLLM achieves optimal FLOP and memory-transfer numbers but performs poorly on reasoning accuracy due to its static attention structure, making it less suitable for long-form reasoning

Table 4: Results of 1K-, 2K-, 4K-, and 6K-token-budget with sampling 16 answers by problem evaluated on Qwen3-4B, Qwen3-8B, and Qwen3-14B, and DeepSeek-R1-Distill-Llama-8B for GPQA benchmarks. For TidalDecode and LessIsMore, we show 16-sample pass@1 as new accuracy (Δ vs. 8-sample in Table 1). Full Attention is shown for reference.

| Model (Task) | Method / Budget | K=1000 | K=2000 | K=4000 | K=6000 |
|---|---|---|---|---|---|
| Qwen-3-4B (GPQA) | TidalDecode | 41.32 (+0.35) | 46.91 (-0.37) | 53.72 (+0.31) | 55.14 |
| | **LessIsMore (Ours)** | **56.48 (+0.17)** | **56.23 (-0.33)** | **56.84 (+0.28)** | **56.64** |
| | Full Attention | 56.19 | 56.19 | 56.19 | 56.19 |
| Qwen-3-8B (GPQA) | TidalDecode | 35.11 (+0.27) | 52.14 (-0.25) | 57.96 (+0.39) | 58.71 |
| | **LessIsMore (Ours)** | **58.62 (-0.22)** | **60.65 (-0.21)** | **61.58 (+0.22)** | **61.11** |
| | Full Attention | 60.54 | 60.54 | 60.54 | 60.54 |
| Qwen-3-14B (GPQA) | TidalDecode | 39.76 (+0.29) | 50.22 (-0.31) | 59.41 (+0.26) | 60.35 |
| | **LessIsMore (Ours)** | **63.42 (+0.29)** | **63.61 (-0.28)** | **64.85 (+0.14)** | **65.15** |
| | Full Attention | 64.02 | 64.02 | 64.02 | 64.02 |
| Llama-8B (GPQA) | TidalDecode | 29.18 (+0.22) | 32.02 (-0.14) | 35.31 (+0.27) | 39.77 |
| | **LessIsMore (Ours)** | **35.74 (+0.27)** | **39.11 (-0.28)** | **43.08 (+0.16)** | **43.31** |
| | Full Attention | 42.80 | 42.80 | 42.80 | 42.80 |

Table 5: The variance of AIME-24 accuracy on Qwen3-8B with different sample sizes over 100 passes. The sampled variance of AIME-24/25, MATH500, and GPQA stay minimal (<0.6) with 64, 8, and 16 samples per problem.

| Model (Task) | Task / Sample Size | 8 | 16 | 64 |
|---|---|---|---|---|
| Qwen-3-8B (Variance) | AIME-24 | ±1.54 | ±0.98 | ±0.56 |
| | AIME-25 | ±1.76 | ±1.12 | ±0.58 |
| | MATH500 | ±0.14 | ±0.11 | ±0.05 |
| | GPQA | ±0.59 | ±0.43 | ±0.20 |

tasks. The full-attention baseline incurs substantially higher FLOPs and memory movement, resulting in much higher latency.

## A.4 CHOICE OF TIDALDECODE OPTIMAL RE-SELECTION ON QWEN3

Following the procedure of choosing optimal re-selection layer of TidalDecode (Yang et al., 2024), we conduct a simple 5K-context-length needle-in-the-haystack test on TidalDecode with each evaluated model. With a token budget of 256, Layer 12 on Qwen3-8B/14B and DeepSeek-R1-Distill-Llama-8B provides the highest accuracy while Layer 12 and Layer 20 on Qwen-4B offer similar performance. Moreover, prior work has found that in the same model family, the optimal re-selection layer is similar. For Qwen3, we validate that Layer 12 is an important layer. To demonstrate the generalization of our approach on different models, we choose different re-selection layers for different Qwen3 models. In this paper's experiments Section 4, we apply the same re-selection layer on TidalDecode and LessIsMore for a fair comparison - Layer 12 and Layer 20 for Qwen3-8B/14B/DeepSeek-R1-Distill-Llama-8B and Qwen3-4B, respectively. We provide a table Table 11 summarizing the re-selection layer used for each model in this paper.

As shown in Table 12, the choice of re-selection layer has a clear and measurable impact on accuracy across all token budgets. Earlier or later layers (e.g., L5, L18, L30) consistently underperform compared to L12, indicating that re-selection must occur at a layer that balances sufficient semantic abstraction with stable attention patterns. LessIsMore+L12 achieves the highest accuracy in every budget setting, matching or exceeding the Full Attention baseline. These results confirm two key points: (1) the re-selection layer is indeed critical for sparse attention performance, and (2) the needle-in-the-haystack search used to identify the optimal TidalDecode layer (L12 for Qwen3-8B) reliably predicts the best re-selection position for LessIsMore as well. This validates our use of the

Table 6: Results of 10K-, 32K-, and 100K-context Needle-in-the-Haystack tests on non-reasoning models Llama-3-8B-Instruct-Gradient-1048k (AI, 2024a) and Llama-3.1-8B-Instruct (AI, 2024b) using the PG-19-mini dataset (Rae et al., 2019). Across both models, LessIsMore consistently matches or surpasses TidalDecode and Quest, demonstrating that its unified token selection effectively captures key information even in purely long-context retrieval settings. Notably, LessIsMore achieves full accuracy with only 32, 32, and 128 tokens for 10K-, 32K-, and 100K-context tasks, corresponding to just 0.3%, 0.1%, and 0.1% of the total input lengths, respectively.

| Model (context length) | Method / Budget | K=32 | K=64 | K=128 | K=256 | K=512 |
|---|---|---|---|---|---|---|
| Llama-3-8B (10K) | Quest | 74% | 84% | 99% | 98% | **100%** |
| | TidalDecode | 88% | 98% | **100%** | **100%** | **100%** |
| | LessIsMore (Ours) | **100%** | **100%** | **100%** | **100%** | **100%** |
| Llama-3-8B (100K) | Quest | 38% | 50% | 65% | 87% | 98% |
| | TidalDecode | 86% | 92% | **100%** | **100%** | **100%** |
| | LessIsMore (Ours) | **98%** | **100%** | **100%** | **100%** | **100%** |
| Llama-3.1-8B (10K) | Quest | 74% | 86% | 94% | **100%** | 98% |
| | TidalDecode | **100%** | **100%** | **100%** | **100%** | **100%** |
| | LessIsMore (Ours) | **100%** | **100%** | **100%** | **100%** | **100%** |
| Llama-3.1-8B (32K) | Quest | 78% | 88% | 92% | **100%** | **100%** |
| | TidalDecode | **98%** | **100%** | **100%** | **100%** | **100%** |
| | LessIsMore (Ours) | **100%** | **100%** | **100%** | **100%** | **100%** |

Table 7: Performance comparison on five LongBench datasets MultiFieldQA(MFQA), Qasper(Qasp), HotpotQA(HotQA), TriviaQA(TrQA), PassageRetrieval-en(PRe), testing capabilities of long-context retrieval, multi-hop Q&A, multi-document comprehension, and structured information integration of each approach. The highest F1-score or accuracy for each task is in bold.

| Method (K) | MFQA | Qasp | HotQA | TrQA | PRe | Avg |
|---|---|---|---|---|---|---|
| Full Attention | 30.76 | **14.56** | 11.50 | 86.56 | 77.00 | 44.08 |
| Quest (1024) | 26.21 | 12.19 | 10.75 | 83.47 | 63.84 | 39.29 |
| TidalDecode (1024) | 28.57 | 11.11 | 9.82 | 79.78 | 75.17 | 40.89 |
| LessIsMore (Ours) (1024) | 29.87 | 14.20 | 12.04 | **87.42** | 75.58 | 43.82 |
| Quest (4096) | 28.92 | 13.63 | 12.15 | 85.91 | 72.50 | 42.62 |
| TidalDecode (4096) | **30.94** | 13.85 | **13.71** | 86.30 | 78.00 | 44.56 |
| LessIsMore (Ours) (4096) | 30.90 | 14.34 | 12.58 | 87.06 | **79.00** | **44.78** |

same re-selection layer for TidalDecode and LessIsMore to ensure a fair and methodologically sound comparison.

## A.5 ABLATION STUDY

### A.5.1 EFFECTIVENESS OF LESSISMORE'S AGGREGATION ON GQA

Applying sparse attention to GQA-based models often requires aggregating tokens from the query heads to the KV group (Yuan et al., 2025). To show the generalizability of our unified selection in LessIsMore beyond the TidalDecode pipeline, we evaluate different token aggregation schemes on Qwen3-8B with GQA, where each KV head is shared across multiple query (or attention) heads. We compare three selection strategies LessIsMore, Randomized Top-k, and Head-to-Head in Figure 6.

The results reveal a critical distinction between local and global optimization strategies. When selection is performed on all decoding layers (All), locally optimal schemes like Randomized Top-k can achieve competitive performance by overfitting to layer-specific patterns. However, when selection frequency is reduced to only Layer 2 (L2)—a more general scenario that reduces computational overhead—our unified approach significantly outperforms other aggregation schemes.

Table 8: End-to-end TBT speed-up of LessIsMore on SGLang serving stack under different context lengths.

| Method | 16K | 32K | 64K |
|---|---|---|---|
| SGLang + LessIsMore-2K | 1.11 | 1.25 | 1.51 |
| SGLang + LessIsMore-4K | 1.09 | 1.22 | 1.48 |

Table 9: End-to-end single-step decoding latency (in ms) with a 2K token budget on DeepSeek-R1-Distilled-LLaMA-8B using the SGLang + FlashInfer serving stack.

| Method (2K) | 16K | 32K | 64K |
|---|---|---|---|
| LessIsMore (Ours) | 23.0ms | 23.4ms | 24.1ms |
| TidalDecode | 24.3ms | 24.7ms | 25.4ms |
| Quest | 24.2ms | 24.4ms | 24.8ms |
| Baseline (Full Attention) | 25.3ms | 28.4ms | 34.4ms |

This performance gap demonstrates that locally optimal selection methods, while effective for immediate layer optimization, fail to generalize robustly across future decoding layers. The superior attention recall of LessIsMore under sparse selection conditions indicates that our global aggregation strategy, combined with the stable recency window, provides more robust token importance estimation that generalizes effectively across the reasoning process. This finding validates the core principle that unified global selection, rather than head-specific local optimization, is essential for maintaining high attention recall in computation-constrained reasoning scenarios.

### A.5.2 GENERATION LENGTH ANALYSIS UNDER SPARSE ATTENTION

Sparse attention methods exhibit a concerning tendency that extends generation lengths on reasoning tasks, as demonstrated in Table 13 and corroborated by prior research (Gao et al., 2025). This phenomenon reflects the accumulation of selection errors discussed in Section 1, where imprecise token retention forces models into inefficient reasoning patterns that compromise both accuracy and computational efficiency.

Table 13 presents the average generation lengths of different approaches under various token budgets on AIME-24 using Qwen3-8B. Under restrictive token budgets (K=2000), existing methods generate substantially longer sequences compared to full attention: Quest, SeerAttention-r and TidalDecode each generate 30.0K, 19.8K, and 17.4K tokens, representing 103%, 34%, and 18% increases respectively over the full attention baseline of 14.8K tokens. These extended sequences indicate that sparse attention errors accumulate over time and may force models to engage in a redundant reasoning process. In contrast, LessIsMore maintains generation lengths closely aligned with full attention across all token budgets. At K=4000, LessIsMore generates the same number of tokens as full attention does while achieving better accuracy. Meanwhile, even with a token budget of 6K, TidalDecode obtains a significant lower accuracy and generates 15.9K tokens. Combining with the average decoding latency in Figure 5a, LessIsMore achieves a $1.16\times$ end-to-end speedup compared to TidalDecode.

Since inaccurate token selection leads to extended generation lengths, attention recall serves as an indicator of both selection accuracy and computational efficiency. Therefore, evaluating attention recall dynamics throughout generation becomes more crucial for assessing sparse attention methods on reasoning tasks.

### A.5.3 EFFECT OF RECENT WINDOW RATIO

To better understand the design choices in LessIsMore, we evaluate how varying the ratio of recent window size impacts attention recall and answer correctness in Figure 7. Curves are annotated with "(T)" or "(F)" to indicate whether the configuration yields the correct answer. Notably, only

Table 10: Kernel-level FLOP count, global-to-shared (G2S) memory transfer, on-device memory consumption, and per-kernel latency for sparse attention with a 2K token budget and 16K context length on DeepSeek-R1-Distilled-LLaMA-8B.

| Method | FLOPs | G2S Transfer | Memory | Latency |
|---|---|---|---|---|
| LessIsMore (Ours) | 1,050,624 | 1.04MB | 8.38MB | 20.08 $\mu s$ |
| TidalDecode | 1,050,624 | 2.34MB | 8.38MB | 32.12 $\mu s$ |
| Quest / SeerAttention-R | 1,050,624 | 2.34MB | 8.38MB | 32.12 $\mu s$ |
| StreamingLLM | 1,050,624 | 1.04MB | 1.04MB | 20.08 $\mu s$ |
| Baseline (Full Attention) | 8,404,992 | 8.38MB | 8.38MB | 76.38 $\mu s$ |

Table 11: Re-selection layers used during evaluation, with model family and task type annotations.

| Model | Family | Attn Type | Reasoning | Re-selection Layer |
|---|---|---|---|---|
| LongChat-7B-v1.5-32k | Llama2 | MHA | ✗ | 7 |
| Llama-3-8B | Llama3 | GQA | ✗ | 13 |
| Llama-3.1-8B | Llama3 | GQA | ✗ | 13 |
| DeepSeek-R1-Distill-Llama-8B | Llama3 | GQA | ✓ | 12 |
| Qwen3-4B | Qwen3 | GQA | ✓ | 20 |
| Qwen3-8B | Qwen3 | GQA | ✓ | 12 |
| Qwen3-14B | Qwen3 | GQA | ✓ | 12 |

configurations that incorporate recent window with Cross-Head Selection (25%, 50%, 75%) succeed in solving the task. We run the AIME-24 reasoning task under a fixed 4K token budget and record the cumulative attention recall across different configurations. Using only recent tokens (100% recent) provides the lowest attention recall, as it discards distant but important contextual tokens. TidalDecode, which selectively retains tokens without explicitly accounting for reasoning-specific attention locality, significantly improves attention recall but still fails to produce the correct answer. Building upon TidalDecode, simply using Cross-Head Selection variant (0% recent) further improves attention recall by leveraging our selection scheme, yet it also fails to generate the correct answer. Incorporating a proportion of recent tokens consistently boosts attention recall. Specifically, configurations with 25%, 50%, and 75% recent windows all manage to generate the correct answer. Among them, 25% recent—which corresponds to the full design of LessIsMore—achieves the highest attention recall throughout the most generation process. This validates the design choice of allocating 25% of the token budget to the recent window in LessIsMore.

### A.5.4 GENERALIZATION OF LESSISMORE ON MHA

LessIsMore is not limited to GQA-based architectures; its unified cross-head token selection strategy naturally extends to standard MHA-based models as well. To demonstrate this generality, we apply LessIsMore to LongChat-7B-v1.5-32k (Li et al., 2023)—an MHA-based long-context model—and evaluate performance on the 10k-context Needle-in-the-Haystack benchmark. As shown in Table 14, LessIsMore consistently matches or surpasses strong selection-based baselines (Quest, TidalDecode) and significantly outperforms eviction-based approaches, achieving full accuracy with only a 256-token budget. These results highlight that LessIsMore captures global token-importance patterns that remain effective even without GQA structure, underscoring its robustness and architectural generality.

### A.6 RECENCY LOCALITY IN REASONING PROCESS

Figure 8 and Figure 9 depicts the ground-truth distribution of selected tokens in GPQA and AIME-25 datasets with the recency locality across attention heads being highlighted. The recency locality is a pattern persistent throughout the thinking process of LRMs regardless of token budgets and tasks. Notably, the size of the highlighted range stays relatively consistent as the model generates more

Table 12: Results of 2K-, 4K-, 6K-, and 8K-token-budget with sampling 64 answers per problem in AIME-24 evaluated on Qwen3-8B. LessIsMore+Lx represents using Layer x as re-selection layer. Full Attention is shown for reference.

| Model (Task) | Method / Budget | K=2000 | K=4000 | K=6000 | K=8000 |
|---|---|---|---|---|---|
| Qwen-3-8B (AIME-24) | LessIsMore+(None) | 58.02 | 63.33 | 66.17 | 70.00 |
| | LessIsMore+L5 | 53.33 | 62.60 | 70.31 | 74.68 |
| | LessIsMore+L18 | 71.67 | 72.91 | 73.33 | 74.79 |
| | LessIsMore+L30 | 63.23 | 65.83 | 70.10 | 75.27 |
| | **LessIsMore+L12 (Ours)** | **73.00** | **75.56** | **76.45** | **76.67** |
| | Full Attention | 74.48 | 74.48 | 74.48 | 74.48 |

Table 13: The AIME-24 accuracy followed by corresponding average reasoning length (in K) of different approaches on Qwen3-8B. The highest accuracy and the lowest generation length of each column are in bold, excluding the Full Attention row.

| Model (Task) | Method / Budget | K=2000 | K=4000 | K=6000 |
|---|---|---|---|---|
| Qwen-3-8B (AIME-24) | Quest | 18.15 (30.0) | 46.67 (22.9) | 49.63 (19.6) |
| | TidalDecode | 53.33 (17.4) | 70.00 (16.9) | 71.30 (15.9) |
| | SeerAttention-r | 58.23 (19.8) | 71.35 (16.3) | 74.06 (15.3) |
| | **LessIsMore (Ours)** | **73.75 (15.8)** | **75.83 (14.8)** | **76.67 (15.1)** |
| | Full Attention | 74.48 (14.8) | 74.48 (14.8) | 74.48 (14.8) |

tokens but grows proportionally to the token budgets. This reinforces the effectiveness of design choice of LessIsMore. Stable Recency Window in Section 4.4.2 leverages the nature of reasoning process and captures the most critical tokens by allocating a fixed ratio of token budgets for most recent tokens.

Table 14: Results of the 10k-context Needle-in-the-Haystack test on the MHA-based LongChat-7B-v1.5-32k model (Li et al., 2023). This experiment demonstrates that LessIsMore generalizes beyond GQA-based architectures, achieving equal or superior accuracy compared to selection-based baselines such as Quest (Tang et al., 2024) and TidalDecode (Yang et al., 2024), and substantially outperforming eviction-based approaches including H2O (Zhang et al., 2024), TOVA (Oren et al., 2024), and StreamingLLM (Xiao et al., 2023). Notably, LessIsMore reaches full accuracy with only a 256-token budget, matching full-attention performance under extreme sparsity.

| Method / Budget | K=32 | K=64 | K=128 | K=256 | K=512 |
|---|---|---|---|---|---|
| H2O | 0% | 1% | 1% | 1% | 3% |
| TOVA | 0% | 1% | 1% | 3% | 8% |
| StreamingLLM | 1% | 1% | 1% | 3% | 5% |
| Quest | 65% | **99%** | **99%** | 99% | **100%** |
| TidalDecode | 73% | 92% | 98% | 99% | **100%** |
| LessIsMore (Ours) | **92%** | 98% | 98% | **100%** | **100%** |

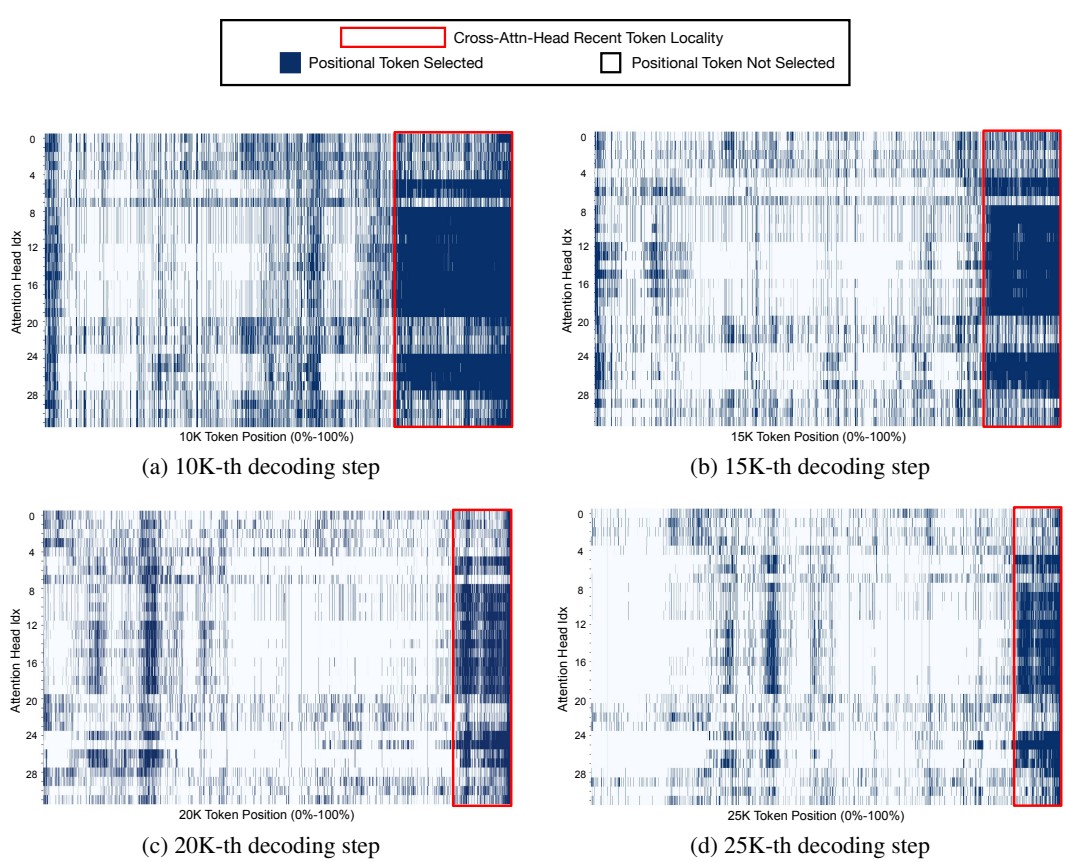

(a) 10K-th decoding step

(b) 15K-th decoding step

(c) 20K-th decoding step

(d) 25K-th decoding step

Figure 8: The distribution of the ground-truth top-4K tokens across all attention heads at 10K-, 15K-, 20K-, and 25K-th decoding step at Layer 4 on AIME-24 with Qwen3-8B. We enclose the highly overlapped area of attention heads within the same KV group with red, which forms a most recent window across all decoding steps

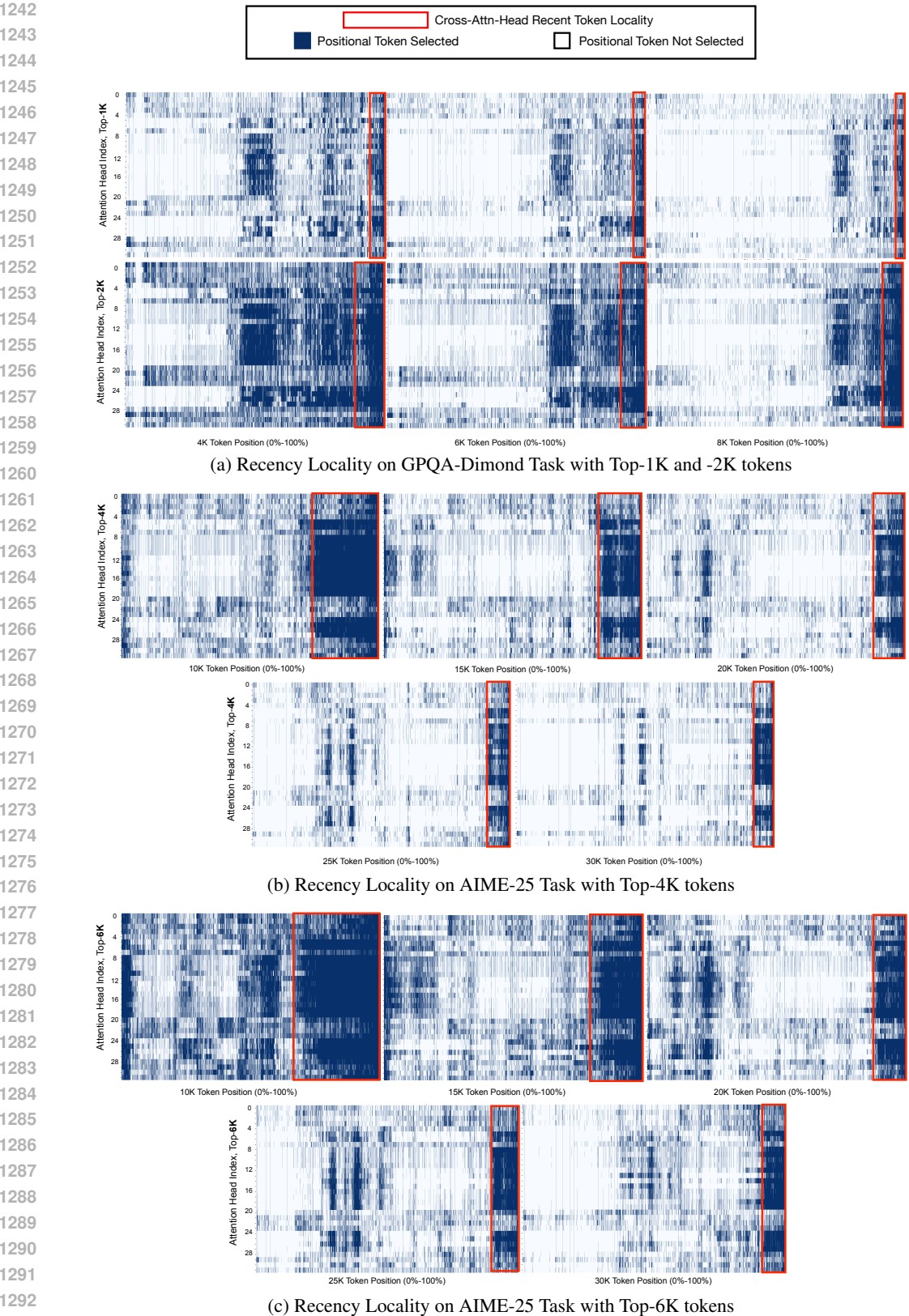

(a) Recency Locality on GPQA-Dimond Task with Top-1K and -2K tokens

(b) Recency Locality on AIME-25 Task with Top-4K tokens

(c) Recency Locality on AIME-25 Task with Top-6K tokens

Figure 9: The ground-truth distribution of selected tokens across different decoding steps and various reasoning tasks on the Layer 4 of Qwen3-8B model.

