# OpenReview forum: "Less Is More: Fast and Accurate Reasoning with Cross-Head Unified Sparse Attention"
_ICLR.cc/2026/Conference — Submitted to ICLR 2026_

### Official Review · Reviewer_Gm7y · 2025-10-26

**Soundness:** 3
**Presentation:** 3
**Contribution:** 3
**Rating:** 6
**Confidence:** 3

**Summary:**

This paper primarily introduces LessIsMore: a post-training and training-free mechanism to speed up the transformer forward pass. The LessIsMore framework targets the attention mechanism, a traditional memory and time bottleneck, in transformer-based LMs. This paper identifies the trend of spatial locality: the observation that there is substantial overlap in token-importance rankings across heads in a given decoding layer. They also identify the trend of recency locality: the observation that recently generated tokens are attended to more highly in subsequent steps. Using the observations of spatial and recency locality, they formulate cross-head unified sparse attention which combines attention head-specific local information with a cross-head global attention pattern to more accurately target relevant tokens during the decoding process. They profile the LessIsMore sparse attention method across several LLMs on several long-context reasoning tasks and show substantial speed-ups with minimal loss in task accuracy.

**Strengths:**

1. This paper tackles an important problem: inference efficiency. With both the size of LLMs and the tasks they are used for increasing, inference can become bottlenecked by the costly attention operation. Further research to enable faster inference with minimal impact to inference accuracy is a promising research area.
2. This paper is well written and easy to follow.
3. The observation of spatial locality is an interesting one; the authors are correct to point out that many prior works have shown that attention heads play “specialized” roles. This paper argues the contrary, and is able to defend their observation by showing that employing a “global” view of token distributions across heads in a given layer informs better token selection for their sparse attention scheme.
4. The adaptive scheme of reserving X% of token positions for the recency window is a logical method to strike the balance between accounting for both recently decoded tokens and earlier tokens (e.g., attention sink tokens [2]).

[1] Lost in the Middle: How Language Models Use Long Contexts. TACL, 2023.

[2] Efficient Streaming Language Models with Attention Sinks. ICLR, 2024.

**Weaknesses:**

- In figures 5.a and 5.b why don’t you compare all methods in each subplot. I want to see the speedup + latency of proposed vs. all baseline methods.
- You reference StreemingLLM [2] in section 2.2.2 as you motivate the design of adaptively sampling tokens from a recency window. Is there a reason you do not include the StreamingLLM decoding method as a baseline strategy in your analysis alongside TidalDecode, SeerAttention-r, Quest?

**Questions:**

- You discuss the time speed-up/latency improvements achieved by LessIsMore. In addition to time, you also cite memory as a motivating concern for the design of LessIsMore in the introduction. Can you discuss the memory footprint of LessIsMore in comparison to your comparison baselines: TidalDecode, SeerAttention-r, Quest?
- Prior work has shown that models often struggle to complete tasks that require use of the “middle” of the context [1]. By design, LessIsMore accounts for more recently decoded tokens, and as you note prior work has shown the presence of “attention sinks” where initially produced tokens hold a high amount of attention mass [2]. Due to LessIsMore’s design of explicitly accounting for more recently generated tokens, and selecting “important” tokens (which may lie at the beginning of a sequence), do you notice that tokens in the “middle” of generation are consistently not selected? Could you discuss LessIsMore in the context of [1]?
- I am not sure if I fully understand what Figure 2 is showing. How do you determine what the “ground-truth top-4K tokens” are? What do you mean by "positional token” (in the legend)? And consequently, what does it mean for a “positional token” to be selected? Some additional text in section 2.2.1/2.2.2 to explain may be helpful.

[1] Lost in the Middle: How Language Models Use Long Contexts. TACL, 2023.

[2] Efficient Streaming Language Models with Attention Sinks. ICLR, 2024.

[1] Lost in the Middle: How Language Models Use Long Contexts. TACL, 2023.

[2] Efficient Streaming Language Models with Attention Sinks. ICLR, 2024.

---

> ### Author Response · Authors · 2025-11-21
> **Reply to Reviewer Gm7y [1/3]**
>
> We sincerely thank reviewer for the valuable feedback.
>
> **TL;DR:** We have conducted comprehensive end-to-end efficiency comparison added across all baselines with SGLang/FlashInfer (Table 9). Memory footprint analysis provided: FLOPs, global-to-shared data transfer, memory consumption (**Table 9 in A1**). StreamingLLM baseline added to all tasks—confirms poor reasoning performance due to "lost in middle" problem, while LessIsMore's design explicitly addresses this via middle token selection (75% budget) + recency window (25% budget). Long-context evaluations on NiTH and LongBench validate effectiveness (**Tables 6–7, 14**). **Figure 2** clarification: shows ground-truth top-4K token positions across query heads in 20K context, demonstrating both recency and cross-head overlap patterns that motivate LessIsMore design.
>
>
> > **W1:** In figures 5.a and 5.b why don't you compare all methods in each subplot. I want to see the speedup + latency of proposed vs. all baseline methods.
>
> **A1**: We agree with the reviewer that an end-to-end efficiency comparison with other sparse attention methods is indeed important and thus add the comparison table below (**Table 9 in paper**). We evaluated the latency for the DeepSeek-R1-Distilled-LLaMA-8B model on a single A5000, so the numbers might differ a bit from the original figure. For each baseline sparse attention method, we are assuming a token budget of 2K, and a context length of 16/32/64K we report a single decoding step's latency with the state-of-the-art serving engine SGLang and FlashInfer as the backend attention kernel library. We can see that our method can consistently achieve lower decoding latency due to a kernel-friendly design (cross-head aggregation) for the GQA-based models. For SeerAttention-R, there are no publicly available end-to-end implementations, but its efficiency should be similar to Quest.
>
> | Method (2K) | 16K | 32K | 64K |
> |---|---|---|---|
> | LessIsMore (Ours) | 23.0ms | 23.4ms | 24.1ms |
> | TidalDecode | 24.3ms | 24.7ms | 25.4ms |
> | Quest | 24.2ms | 24.4ms | 24.8ms |
> | Baseline (Full Attention) | 25.3ms | 28.4ms | 34.4ms |
>
> **Table 9 Caption:** End-to-end single-step decoding latency (in ms) with a 2K token budget on DeepSeek-R1-Distilled-LLaMA-8B using the SGLang + FlashInfer serving stack.
>
> > **W2:** You reference StreemingLLM [2] in section 2.2.2 as you motivate the design of adaptively sampling tokens from a recency window. Is there a reason you do not include the StreamingLLM decoding method as a baseline strategy in your analysis alongside TidalDecode, SeerAttention-r, Quest?
>
> **A2**: We thank the reviewer for mentioning adding StreamingLLM as a baseline. In **Figure 4** in 4.2, we can clearly see that StreamingLLM constantly achieves the worst performance due to the fact that it discards tokens in the middle, which exactly corroborates with the Lost in the Middle work mentioned by the reviewer. In addition, we have further evaluated LessIsMore and sparse attention baselines on long context tasks, including needle-in-the-haystack and LongBench testing its long-context capabilities, ranging from retrieval, multi-hop Q&A, multi-document comprehension, and long context understanding. Evaluation results in **Table 6, 7, and 14 (all attached below)** in A.2 show that across various token budgets and datasets, LessIsMore achieves close accuracy to the full attention baseline and better results than other sparse attention approaches.
>
> | Method (K) | MultiFieldQA | Qasper | HotpotQA | TriviaQA | PassageRetrieval-en | Avg |
> |---|---|---|---|---|---|---|
> | Full Attention | 30.76 | **14.56** | 11.50 | 86.56 | 77.00 | 44.08 |
> | Quest (1024) | 26.21 | 12.19 | 10.75 | 83.47 | 63.84 | 39.29 |
> | TidalDecode (1024) | 28.57 | 11.11 | 9.82 | 79.78 | 75.17 | 40.89 |
> | LessIsMore (Ours) (1024) | 29.87 | 14.20 | 12.04 | **87.42** | 75.58 | 43.82 |
> | Quest (4096) | 28.92 | 13.63 | 12.15 | 85.91 | 72.50 | 42.62 |
> | TidalDecode (4096) | **30.94** | 13.85 | **13.71** | 86.30 | 78.00 | 44.56 |
> | LessIsMore (Ours) (4096) | 30.90 | 14.34 | 12.58 | 87.06 | **79.00** | **44.78** |
>
> **Table 7 Caption:** Performance comparison on five LongBench datasets MultiFieldQA, Qasper, HotpotQA, TriviaQA, and PassageRetrieval-en of each approach. The highest F1-score or accuracy for each task is in bold.

---

> ### Author Response · Authors · 2025-11-21
> **Reply to Reviewer Gm7y [2/3]**
>
> | Model (context length) | Method / Budget | K=32 | K=64 | K=128 | K=256 | K=512 |
> |---|---|---|---|---|---|---|
> | Llama-3-8B (10K) | Quest | 74% | 84% | 99% | 98% | **100%** |
> | | TidalDecode | 88% | 98% | **100%** | **100%** | **100%** |
> | | LessIsMore (Ours) | **100%** | **100%** | **100%** | **100%** | **100%** |
> | Llama-3-8B (100K) | Quest | 38% | 50% | 65% | 87% | 98% |
> | | TidalDecode | 86% | 92% | **100%** | **100%** | **100%** |
> | | LessIsMore (Ours) | **98%** | **100%** | **100%** | **100%** | **100%** |
> | Llama-3.1-8B (10K) | Quest | 74% | 86% | 94% | **100%** | 98% |
> | | TidalDecode | **100%** | **100%** | **100%** | **100%** | **100%** |
> | | LessIsMore (Ours) | **100%** | **100%** | **100%** | **100%** | **100%** |
> | Llama-3.1-8B (32K) | Quest | 78% | 88% | 92% | **100%** | **100%** |
> | | TidalDecode | **98%** | **100%** | **100%** | **100%** | **100%** |
> | | LessIsMore (Ours) | **100%** | **100%** | **100%** | **100%** | **100%** |
>
> **Table 6 Caption:** Results of 10K-, 32K-, and 100K-context NiTH on non-reasoning models Llama-3-8B-Instruct-Gradient-1048k and Llama-3.1-8B-Instruct using the PG-19-mini dataset.
>
> | Method / Budget | K=32 | K=64 | K=128 | K=256 | K=512 |
> |---|---|---|---|---|---|
> | H2O | 0% | 1% | 1% | 1% | 3% |
> | TOVA | 0% | 1% | 1% | 3% | 8% |
> | StreamingLLM | 1% | 1% | 1% | 3% | 5% |
> | Quest | 65% | **99%** | **99%** | 99% | **100%** |
> | TidalDecode | 73% | 92% | 98% | 99% | **100%** |
> | LessIsMore (Ours) | **92%** | 98% | 98% | **100%** | **100%** |
>
> **Table 14 Caption:** Results of the 10k-context Needle-in-the-Haystack test on the MHA-based LongChat-7B-v1.5-32k model. LessIsMore achieves equal or superior accuracy compared to selection-based baselines such as Quest[1] and TidalDecode[2], and substantially outperforming eviction-based approaches including H2O[2], TOVA[3], and StreamingLLM[5].
>
> > **Q1:** You discuss the time speed-up/latency improvements achieved by LessIsMore. In addition to time, you also cite memory as a motivating concern for the design of LessIsMore in the introduction. Can you discuss the memory footprint of LessIsMore in comparison to your comparison baselines: TidalDecode, SeerAttention-r, Quest?
>
> **A3**: Below in **Table 10** we show the FLOP/Global-to-Shared memory data transfer/Memory consumption in global memory and per kernel latency value with a sparse token budget of 2K and context length of 16K with DeepSeek-R1-Distilled-LLaMA-8B. We can clearly see that LessIsMore can achieve less data transfer when comparing against TidalDecode and Quest, which is the core reason for our kernel speed-up. Even though StreamingLLM achieves the optimal numbers under all metrics, it achieves the worst performance on reasoning accuracy, as shown in **Figure 4**. In practice, Quest and SeerAttention-R require a bit more device memory due to their auxiliary data for efficient token importance estimation.
>
> | Method | FLOPs | G2S Transfer | Memory | Latency |
> |---|---|---|---|---|
> | LessIsMore (Ours) | 1,050,624 | 1.04MB | 8.38MB | 20.08 μs |
> | TidalDecode | 1,050,624 | 2.34MB | 8.38MB | 32.12 μs |
> | Quest / SeerAttention-R | 1,050,624 | 2.34MB | 8.38MB | 32.12 μs |
> | StreamingLLM | 1,050,624 | 1.04MB | 1.04MB | 20.08 μs |
> | Baseline (Full Attention) | 8,404,992 | 8.38MB | 8.38MB | 76.38 μs |
>
> **Table 10 Caption:** Kernel-level FLOP count, global-to-shared (G2S) memory transfer, on-device memory consumption, and per-kernel latency for sparse attention with a 2K token budget and 16K context length on DeepSeek-R1-Distilled-LLaMA-8B.
>
> > **Q2:** Prior work has shown that models often struggle to complete tasks that require use of the "middle" of the context [1]. By design, LessIsMore accounts for more recently decoded tokens, and as you note prior work has shown the presence of "attention sinks" where initially produced tokens hold a high amount of attention mass [2]. Due to LessIsMore's design of explicitly accounting for more recently generated tokens, and selecting "important" tokens (which may lie at the beginning of a sequence), do you notice that tokens in the "middle" of generation are consistently not selected? Could you discuss LessIsMore in the context of [1]?
>
> **A4**: We totally agree with the reviewer that Lost in the Middle is indeed a crucial problem for eviction-based sparse attention algorithms like StreamingLLM or H2O. That is why in our design, besides the sink tokens and the most recent window tokens that take up 25% of the overall token budget, we need to have the selection layers to pick the remaining 75% tokens in the middle. And as shown in our updated experiments in **Figure 4** in 4.2 and ablation study in **Figure 7** in 4.4, doing so can largely improve model performance on complicated tasks like reasoning, while methods like StreamingLLM achieve worse performance.

---

> ### Author Response · Authors · 2025-11-21
> **Reply to Reviewer Gm7y [3/3]**
>
> > **Q3:** I am not sure if I fully understand what Figure 2 is showing. How do you determine what the "ground-truth top-4K tokens" are? What do you mean by "positional token" (in the legend)? And consequently, what does it mean for a "positional token" to be selected? Some additional text in section 2.2.1/2.2.2 to explain may be helpful.
>
> **A5**: Sorry for the confusion of **Figure 2**. We have updated our descriptions accordingly in the revised paper. So the X-axis of Figure 2 is the token's relative position in one decoding step, where the sequence length is 20K, so the token position ids will be 0-20K from left to right. The Y-axis shows different query heads, and for each query head's query vector, we get the ground truth top-4K tokens by examining its corresponding attention score over all 20K tokens and highlighting 4K tokens with the highest attention scores. The positional token is just any token position (any token index along the X-axis) within the 20K past tokens. Selected means it lies in the ground truth top-4K token set, and we highlight these selected tokens in dark blue. We mainly want to show the recency pattern as well as the cross-head patterns in the middle to motivate our token selection in the middle + recency window design.

---

> > ### Comment · Reviewer_Gm7y · 2025-11-26
> >
> > Thank you for the clarification and the extra experiments. I do not have additional questions and would like to maintain my current positive rating.

---

### Official Review · Reviewer_nhSJ · 2025-10-31

**Soundness:** 3
**Presentation:** 3
**Contribution:** 2
**Rating:** 4
**Confidence:** 4

**Summary:**

Training-free method that induces sparsity in attention via token selection/aggregation across heads during decoding. Evaluated mainly on math reasoning (e.g., AIME-24/25, MATH500, GPQA-Diamond). Shows accuracy vs. token-budget tradeoffs and latency relative to a dense baseline.

**Strengths:**

- Simple, training-free mechanism to enforce attention sparsity.
- Comprehensive math-reasoning evaluation with clear pass@1 reporting.
- Appears drop-in and compatible with standard decoding stacks.

**Weaknesses:**

- The statement on index sharing across layers (Line 296) could use further justification or empirical evidence — for example, analyzing whether top-k tokens remain stable across layers in practice.
- The main novelty appears to lie in token aggregation across attention heads; adding more discussion or ablations could help highlight its unique contribution.
- The evaluation setup (e.g., “average pass@1 over 16/8 samples”) could be elaborated — how were samples selected, and what motivates such small test sets?
- Figure 4 shows performance vs token budget for different SOTA sparse attention methods, however there is no efficiency (flops/latency saved ) analysis for different sparse attention methods , figure 5 shows latency comparison but its only relative to baseline (no sparsity).
- It would be helpful to see how different sparse attention baselines behaves for longer context lengths (beyond 8K/16K tokens).
- Adding more sparse attention baselines (see Efficient Attention Mechanisms for Large Language Models: A Survey, arXiv:2507.19595) could provide better context.
- Since the method targets efficient reasoning, exploring broader tasks—such as code generation, tool-use, or long-context reasoning (LongBench, needle-in-a-haystack)—would test its generality and robustness.
- The idea of focusing attention on recent tokens has been explored in earlier works (e.g., StreamLLM and related sparse attention methods)

**Questions:**

- What is the rationale for assuming top-k token indices remain useful across subsequent layers?
- Could the authors provide ablations isolating the effects of cross-head aggregation and token selection frequency?
- Can they include absolute compute metrics (FLOPs, latency per sequence, memory usage) for better comparison?
- How does sparsity affect performance on long-context or structured tasks?
- Are there plans to expand baselines to include more diverse sparse attention approaches?

---

> ### Author Response · Authors · 2025-11-21
> **Reply to Reviewer nhSJ [1/4]**
>
> We sincerely thank reviewer for the valuable feedback.
>
> **TL;DR:** We have added StreamingLLM baseline and extended token budgets to 6K(MATH+GPQA)/8K(AIME24+25) with larger sample sizes (**Figure 4**). Increased sample sizes: 64 for AIME, 16 for GPQA across 30/198/500 total problems respectively (Tables 3–4). Explanation on cross-head aggregation ablations is elaborated, showing significant attention recall improvements over single-head approaches (**Figure 6**). End-to-end efficiency metrics provided: FLOPs, latency, memory usage, and data transfer (**Table 10**). Extended evaluations to long-context tasks: NiTH (10K/32K/100K) and LongBench datasets (Tables 6–7, 14) with more baselines.
>
> > **W1:** The statement on index sharing across layers (Line 296) could use further justification or empirical evidence — for example, analyzing whether top-k tokens remain stable across layers in practice.
>
> **A1**: We thank the reviewer for mentioning the relation of index overlapping and accuracy. In Figure 1(a) in 2.1, we showed that the overall attention recall improvement from LessIsMore compared to other sparse attention approaches. Prior work [1] has comprehensively analyzed the high Top-K similarity across adjacent layers within a single decoding step. LessIsMore focuses on leveraging the shared pattern across all attention heads and a fixed ratio of recency window to improve attention recall, which directly impacts accuracy. To further validate the effectiveness of our design, we have added new evals on StreamingLLM[2] for all reasoning tasks and token budgets up to 8K and have plotted new results in Figure 4 in 4.2. Figure 1(a) and Figure 4 collectively show that empirically LessIsMore consistently achieves higher attention recall throughout the generation process than other approaches, which leads it to consistently outperform all other sparse attention baselines and stay close to Full Attention.
>
> > **W2:** The main novelty appears to lie in token aggregation across attention heads; adding more discussion or ablations could help highlight its unique contribution.
>
> **A2**: We agree with the reviewer that token aggregation is indeed important, and thus we had the figure in the ablation study section Figure 6 in 4.3 showing the effectiveness of the token integration strategy in LessIsMore compared to other approaches. Cross-head integration with the recent window in LessIsMore produces significantly higher attention recall compared to random single-head's top-K integration or even locally optimal head-to-head mapping. Moreover, we added new experiments of applying LessIsMore on a non-GQA-based model to test its long-context capabilities in Table 14(attached below). The new experiments demonstrate the competitiveness of LessIsMore beyond reasoning and GQA mechanisms.
>
> | Method / Budget | K=32 | K=64 | K=128 | K=256 | K=512 |
> |---|---|---|---|---|---|
> | H2O | 0% | 1% | 1% | 1% | 3% |
> | TOVA | 0% | 1% | 1% | 3% | 8% |
> | StreamingLLM | 1% | 1% | 1% | 3% | 5% |
> | Quest | 65% | **99%** | **99%** | 99% | **100%** |
> | TidalDecode | 73% | 92% | 98% | 99% | **100%** |
> | LessIsMore (Ours) | **92%** | 98% | 98% | **100%** | **100%** |
>
> **Table 14 Caption:** Results of the 10k-context Needle-in-the-Haystack test on the MHA-based LongChat-7B-v1.5-32k model. LessIsMore achieves equal or superior accuracy compared to selection-based baselines such as Quest[1] and TidalDecode[2], and substantially outperforming eviction-based approaches including H2O[2], TOVA[3], and StreamingLLM[5].
>
> > **W3:** The evaluation setup (e.g. "average pass@1 over 16/8 samples") could be elaborated — how were samples selected, and what motivates such small test sets?
>
> **A3**: We want first to clarify that the sample size number here is not how many problems/requests we have evaluated from the corresponding dataset (AIME-24/25, GPQA/MATH500). We indeed evaluate all the problems in each dataset, which is 30 problems for AIME-24/25, 198 problems in GPQA and 500 problems for MATH500. But since the model generation is commonly stochastic, for each problem we generate more than one response, which we refer to as the sample size. Then we average over all the generated responses to get the averaged accuracy for such a problem. We apologize for any confusion that we introduced by overusing the sample size term, and we have refined our writing accordingly. Under such cases, a sample size of 16 is typically low-variance in **Table 5** (attached below), showing relationships of variance vs. sample size on different tasks. But to further address the reviewer's concern, we increased the sample size to 64 for AIME-24/25 and 16 for GPQA. Final accuracies and differences between new and old ones are shown in **Table 3 & 4** in A.1.2. For all the sample sizes used in our paper and updated ones, LessIsMore always achieves the best accuracy and small variance across all token budgets and achieves a similar accuracy with the full attention baseline with small token budgets.

---

> ### Author Response · Authors · 2025-11-21
> **Reply to Reviewer nhSJ [2/4]**
>
> | Model (Task) | Task / Sample Size | 8 | 16 | 64 |
> |---|---|---|---|---|
> | Qwen-3-8B (Variance) | AIME-24 | ±1.54 | ±0.98 | ±0.56 |
> | | AIME-25 | ±1.76 | ±1.12 | ±0.58 |
> | | MATH500 | ±0.14 | ±0.11 | ±0.05 |
> | | GPQA | ±0.59 | ±0.43 | ±0.20 |
>
> **Table 5 Caption:** The variance of AIME-24 accuracy on Qwen3-8B. The sampled variance of AIME-24/25, MATH500, and GPQA stay minimal (<0.6) with 64, 8, and 16 samples per problem.
>
> > **W4:** Figure 4 shows performance vs token budget for different SOTA sparse attention methods, however there is no efficiency (flops/latency saved) analysis for different sparse attention methods, figure 5 shows latency comparison but its only relative to baseline (no sparsity).
>
> **A4**: We agree with the reviewer that FLOP/Memory/Data transferred/Latency are indeed important metrics to report; we listed them in the table attached below (as paper's **Table 10**). All the numbers are calculated or measured for a single layer's attention kernel. The context length here is 16K, and for all sparse attention methods, we assume a token budget of 2K. We can clearly see that LessIsMore can achieve less global-to-share (G2S) data transfer when comparing against TidalDecode and Quest, which is the core reason for our kernel speed-up. Even though StreamingLLM achieves the optimal numbers under all metrics, it achieves the worst performance on reasoning accuracy, as shown in **Figure 4**.
>
> | Method | FLOPs | G2S Transfer | Memory | Latency |
> |---|---|---|---|---|
> | LessIsMore (Ours) | 1,050,624 | 1.04MB | 8.38MB | 20.08 μs |
> | TidalDecode | 1,050,624 | 2.34MB | 8.38MB | 32.12 μs |
> | Quest / SeerAttention-R | 1,050,624 | 2.34MB | 8.38MB | 32.12 μs |
> | StreamingLLM | 1,050,624 | 1.04MB | 1.04MB | 20.08 μs |
> | Baseline (Full Attention) | 8,404,992 | 8.38MB | 8.38MB | 76.38 μs |
>
> **Table 10 Caption:** Kernel-level FLOP count, global-to-shared (G2S) memory transfer, on-device memory consumption, and per-kernel latency for sparse attention with a 2K token budget and 16K context length on DeepSeek-R1-Distilled-LLaMA-8B.
>
> > **W5:** It would be helpful to see how different sparse attention baselines behaves for longer context lengths (beyond 8K/16K tokens).
>
> **A5**: We thank the reviewer for mentioning performance on larger context limits. Most of the reasoning models we evaluated have a context limit of 32K by default. To evaluate LessIsMore and related baselines on a longer-context, we have added the 10K/32K/100K Needle-in-the-haystack tests in **Table 6** (attached below) and **Table 14**(attached in **A2**), showing LessIsMore outperforming all other sparse attention baselines in long-context settings. For efficiency evaluation, we presented end-to-end efficiency.
>
> | Model (context length) | Method / Budget | K=32 | K=64 | K=128 | K=256 | K=512 |
> |---|---|---|---|---|---|---|
> | Llama-3-8B (10K) | Quest | 74% | 84% | 99% | 98% | **100%** |
> | | TidalDecode | 88% | 98% | **100%** | **100%** | **100%** |
> | | LessIsMore (Ours) | **100%** | **100%** | **100%** | **100%** | **100%** |
> | Llama-3-8B (100K) | Quest | 38% | 50% | 65% | 87% | 98% |
> | | TidalDecode | 86% | 92% | **100%** | **100%** | **100%** |
> | | LessIsMore (Ours) | **98%** | **100%** | **100%** | **100%** | **100%** |
> | Llama-3.1-8B (10K) | Quest | 74% | 86% | 94% | **100%** | 98% |
> | | TidalDecode | **100%** | **100%** | **100%** | **100%** | **100%** |
> | | LessIsMore (Ours) | **100%** | **100%** | **100%** | **100%** | **100%** |
> | Llama-3.1-8B (32K) | Quest | 78% | 88% | 92% | **100%** | **100%** |
> | | TidalDecode | **98%** | **100%** | **100%** | **100%** | **100%** |
> | | LessIsMore (Ours) | **100%** | **100%** | **100%** | **100%** | **100%** |
>
> **Table 6 Caption:** Results of 10K-, 32K-, and 100K-context NiTH on non-reasoning models Llama-3-8B-Instruct-Gradient-1048k and Llama-3.1-8B-Instruct using the PG-19-mini dataset.
>
> > **W6:** Adding more sparse attention baselines (see Efficient Attention Mechanisms for Large Language Models: A Survey, arXiv:2507.19595) could provide better context.
>
> **A6**: LessIsMore is designed to target improving reasoning efficiency, which requires a more accurate selection scheme compared to traditional sparse attention approaches for long-context tasks. We already included two of the most performant sparse attention baselines for long-context tasks, TidalDecode and Quest, and a post-training-required approach specifically for reasoning tasks, Seer-Attention-R, where we already consistently outperform all of them in the original paper. Furthermore, to show the effectiveness of LessIsMore, we have further added reasoning evaluation of StreamingLLM on all reasoning tasks in **Figure 4** and compared LessIsMore with Quest[4]/TidalDecode[1]/TOVA[5]/H2O[6]/StreamingLLM[7] from survey [2] for long-context retrieval tasks in **Table 14 (attached in A2)**. New reasoning and retrieval eval results show that LessIsMore achieves superior accuracy than all evaluated approaches on either reasoning or long-context tasks.

---

> ### Author Response · Authors · 2025-11-21
> **Reply to Reviewer nhSJ [3/4]**
>
> > **W7:** Since the method targets efficient reasoning, exploring broader tasks—such as code generation, tool-use, or long-context reasoning (LongBench, needle-in-a-haystack)—would test its generality and robustness.
>
> **A7**: We have added the evaluation results of LessIsMore on needle-in-the-haystack task and LongBench datasets, testing its long-context capabilities ranging from retrieval, multi-hop Q&A, multi-doc comprehension, and long context understanding. Evaluated results in, **Table 6 (attached in A5)**,  **Table 7** (attached below) , and **Table 14 (attached in A2)** show that across various token budgets and datasets, LessIsMore achieves lossless accuracy to the full attention baseline and better results than other sparse attention approaches, even obtaining a higher average score in LongBench than Full Attention baseline.
>
> | Method (K) | MultiFieldQA | Qasper | HotpotQA | TriviaQA | PassageRetrieval-en | Avg |
> |---|---|---|---|---|---|---|
> | Full Attention | 30.76 | **14.56** | 11.50 | 86.56 | 77.00 | 44.08 |
> | Quest (1024) | 26.21 | 12.19 | 10.75 | 83.47 | 63.84 | 39.29 |
> | TidalDecode (1024) | 28.57 | 11.11 | 9.82 | 79.78 | 75.17 | 40.89 |
> | LessIsMore (Ours) (1024) | 29.87 | 14.20 | 12.04 | **87.42** | 75.58 | 43.82 |
> | Quest (4096) | 28.92 | 13.63 | 12.15 | 85.91 | 72.50 | 42.62 |
> | TidalDecode (4096) | **30.94** | 13.85 | **13.71** | 86.30 | 78.00 | 44.56 |
> | LessIsMore (Ours) (4096) | 30.90 | 14.34 | 12.58 | 87.06 | **79.00** | **44.78** |
>
> **Table 7 Caption:** Performance comparison on five LongBench datasets MultiFieldQA, Qasper, HotpotQA, TriviaQA, and PassageRetrieval-en of each approach. The highest F1-score or accuracy for each task is in bold.
>
> > **W8:** The idea of focusing attention on recent tokens has been explored in earlier works (e.g., StreamLLM and related sparse attention methods)
>
> **A8**: Yes, we have recognized that the recency window concept has been explored in earlier works such as StreamingLLM [2] and DeepSeek NSA [3] papers. They often tend to maintain a fixed size of tokens as a sliding window (such as 256 in NSA, regardless of token budgets). However, as far as we know, LessIsMore is the first work that observes, analyzes, and applies the constant ratio between token budgets and prominent recency window size to reasoning models and tasks. We have performed a comprehensive ablation study in **Figure 7** in 4.3 to reveal its effect on accuracy and plotted heatmaps across different reasoning tasks in **Figure 8 & 9** in A.6 to show the consistency of the recency window. This observation signals that the window size is supposed to shrink proportionally to the token budgets in reasoning to optimally attribute the compute budget. Besides, StreamingLLM alone cannot achieve great reasoning accuracy, as shown in **Figure 4** in 4.2. As far as we know, our work is the first training-free sparse attention method that achieves nearly no accuracy loss on reasoning tasks with only a limited token budget.
>
> > **Q1:** What is the rationale for assuming top-k token indices remain useful across subsequent layers?
>
> **A9**: Prior work TidalDecode [1] first observes that tokens that receive the highest attention scores share positional similarity in adjacent layers. However, the prior work naively applies head-to-head top-k mapping in the selection layer and ignores the overall attention pattern, leading to significant attention recall drop (**Figure 6** in 4.3) and thus accuracy degradation (**Figure 4** in 4.2) in reasoning. Therefore, LessIsMore rectifies this accumulated error by leveraging the observation of cross-head integration and a stable recency window to improve the accuracy and efficiency in reasoning.
>
> > **Q2:** Could the authors provide ablations isolating the effects of cross-head aggregation and token selection frequency?
>
> **A10**: In the ablation study **Figure 6** in 4.3 of the original paper, we have provided the ablation study on attention recall comparing our head-to-head integration and other approaches on a one-layer and every-layer basis. The figure shows that even if LessIsMore provides slightly lower attention recall when selecting and integrating selected top-k tokens for every layer, it achieves significantly higher attention recall than others when reducing the selection to Layer-2-only, showing the effectiveness of cross-head integration in sparse attention for reasoning. Additionally, when comparing against TidalDecode with only two selection layers, we are essentially sharing the same selection frequency but have a different aggregation approach, and the results show that our method can achieve significantly better reasoning performance compared with TidalDecode.
>
> > **Q3:** Can they include absolute compute metrics (FLOPs, latency per sequence, memory usage) for better comparison?
>
> **A11**: Please refer to W4.

---

> ### Author Response · Authors · 2025-11-21
> **Reply to Reviewer nhSJ [4/4]**
>
> > **Q4:** How does sparsity affect performance on long-context or structured tasks?
>
> **A12**: We have evaluated LessIsMore and sparse attention baselines on long context tasks, including needle-in-the-haystack and LongBench testing its long-context capabilities, ranging from retrieval, summarization, multi-hop Q&A, multi-doc comprehension and long context understanding. Evaluation results in **Table 6 & 14 (needle-in-the-haystack, attached in A2 and A5)** and **7 (longbench datasets, attached in A7)** show that across various token budgets and datasets, LessIsMore achieves close accuracy to the full attention baseline and better results than other sparse attention approaches.
>
> > **Q5:** Are there plans to expand baselines to include more diverse sparse attention approaches?
>
> **A13**: Yes, we have already included more baselines such as StreamingLLM, H2O, and TOVA in both reasoning and long-context evaluations. LessIsMore significantly outperforms all of these sparse attention baselines as shown in **Figure 4 in 4.2 (reasoning)**, Table 14 (needle-in-the-haystack attached in A2 and A5), and **Table 7 (LongBench attached in A7)**.
>
> References:
>
> [1]. Yang, Lijie, et al. "Tidaldecode: Fast and accurate llm decoding with position persistent sparse attention." arXiv preprint arXiv:2410.05076 (2024).
>
> [2] Efficient Attention Mechanisms for Large Language Models: A Survey, arXiv:2507.19595
>
> [3] Native Sparse Attention: Hardware-Aligned and Natively Trainable Sparse Attention, https://arxiv.org/abs/2502.11089
>
>
> [4] J Tang, et al. "Quest: Query-Aware Sparsity for Efficient Long-Context LLM Inference." arXiv preprint arXiv:2406.10774, 2024.
>
> [5] Z Zhang, et al. "H₂O: Heavy-Hitter Oracle for Efficient Generative Inference of Large Language Models." arXiv preprint arXiv:2306.14048, 2023.
>
> [6] M Oren, et al. "Transformers are Multi-State RNNs." arXiv preprint arXiv:2401.06104, 2024.
>
> [7] G Xiao, et al. "Efficient Streaming Language Models with Attention Sinks." arXiv preprint arXiv:2309.17453, 2023.

---

### Official Review · Reviewer_K4Av · 2025-11-01

**Soundness:** 2
**Presentation:** 3
**Contribution:** 3
**Rating:** 4
**Confidence:** 2

**Summary:**

This paper proposes "LessIsMore" which is a training-free sparse-attention method tailored for long generation reasoning tasks. Instead of letting each head pick its own top-k tokens, it aggregates head-level candidates into a global, cross-head ranking and always keeps a fixed recency window of the latest tokens. On Qwen3-8B/4B across reasoning benchmarks (AIME-24/25, GPQA-Diamond, MATH500), it reports near-lossless accuracy while attending to about 2× fewer tokens and achieving roughly 1.1× average decoding speedups compared to full attention.

**Strengths:**

1) This is a simple, training-free method which doesn't require model changes or retraining.

2) The paper Identifies spatial locality (overlap across heads) and recency locality (recent tokens remain important), motivating a global and recency selector.

3) They show strong empirical results on reasoning tasks which matches or improves accuracy at higher sparsity; avoids the lengthening of generations that hurts some sparse methods.

**Weaknesses:**

1) Their results only focus on e.g. Qwen3 (with GQA backbone), questions remain how does it work on non-GQA/MHA models (e.g., Llama-3.x, Mistral)?

2) Some gains may be tied to custom GQA kernel support. Can you quantify benefits under other stacks like (vLLM/Flash-Attention) and with speculative decoding or KV-cache quantitation?

3) Is the cross-head overlap is something specialised to GQA or a general property? Can you show comparisons between MHA and GQA by keeping other factors fixed?

**Questions:**

Look at the weaknesses.

---

> ### Author Response · Authors · 2025-11-21
> **Reply to Reviewer K4Av [1/2]**
>
> We sincerely thank reviewer for the valuable feedback.
>
> > **W1:** Their results only focus on e.g. Qwen3 (with GQA backbone), questions remain how does it work on non-GQA/MHA models (e.g., Llama-3.x, Mistral)?
>
> **A1**: We recognize that model generalizability is crucial. In the experiment section Figure 4 in 4.2, we already provided models beyond the Qwen family such as DeepSeek-R1-Distill-Llama-8B. We added more comprehensive new experiments of StreamingLLM and extended token budgets to 8K with larger sample sizes in Figure 4 with values recorded in Table **1**, **2** (attached below) 3, and 4 in A.1.1 and A.1.2. Across all reasoning models, token budgets, and reasoning tasks evaluated, LessIsMore always achieves the same or even better accuracy as the dense model baseline with small token budgets while outperforming all other sparse attention baselines. Albeit we are exploring reasoning models that are mostly GQA-based architectures, to show the generalization of LessIsMore on other attention mechanisms, such as MHA, we compared the performance of LessIsMore on long-context tasks with non-reasoning models in **Table 14** (attached below), showing the generalization to different attention variants of our approach.
>
> | Model (Task) | Method / Budget | K=2000 | K=4000 | K=6000 | K=8000 |
> |---|---|---|---|---|---|
> | DeepSeek-R1-Distill-Llama-8B (AIME-24) | Quest | 16.67 | 31.67 | 35.21 | 41.25 |
> | | TidalDecode | 39.16 | 41.25 | 43.33 | 44.11 |
> | | **LessIsMore (Ours)** | **43.33** | **44.16** | **46.67** | **45.10** |
> | | Full Attention | 44.16 | 44.16 | 44.16 | 44.16 |
> | DeepSeek-R1-Distill-Llama-8B (AIME-25) | Quest | 16.67 | 18.33 | 20.00 | 24.58 |
> | | TidalDecode | 24.97 | 25.14 | 27.71 | 28.33 |
> | | **LessIsMore (Ours)** | **30.42** | **31.04** | **31.25** | **31.67** |
> | | Full Attention | 30.83 | 30.83 | 30.83 | 30.83 |
>
> | Model (Task) | Method / Budget | K=1000 | K=2000 | K=4000 | K=6000 |
> |---|---|---|---|---|---|
> | DeepSeek-R1-Distill-Llama-8B (MATH500) | Quest | 67.75 | 72.78 | 75.99 | 80.08 |
> | | TidalDecode | 77.21 | 80.68 | 82.72 | 84.68 |
> | | **LessIsMore (Ours)** | **86.90** | **88.15** | **88.75** | **88.54** |
> | | Full Attention | 85.45 | 85.45 | 85.45 | 85.45 |
> | DeepSeek-R1-Distill-Llama-8B (GPQA) | Quest | 22.73 | 30.30 | 34.09 | 37.63 |
> | | TidalDecode | 28.96 | 32.16 | 35.04 | 39.77 |
> | | **LessIsMore (Ours)** | **35.47** | **39.39** | **42.92** | **43.31** |
> | | Full Attention | 42.80 | 42.80 | 42.80 | 42.80 |
>
> **Table 1 & 2 Caption:** Results of LessIsMore (Ours), Quest[2], TidalDecode[3] on Llama model that are originally in paper. We extended AIME-24/25 to 8K and MATH/GPQA 6K token budgets.
>
> | Method / Budget | K=32 | K=64 | K=128 | K=256 | K=512 |
> |---|---|---|---|---|---|
> | H2O | 0% | 1% | 1% | 1% | 3% |
> | TOVA | 0% | 1% | 1% | 3% | 8% |
> | StreamingLLM | 1% | 1% | 1% | 3% | 5% |
> | Quest | 65% | **99%** | **99%** | 99% | **100%** |
> | TidalDecode | 73% | 92% | 98% | 99% | **100%** |
> | LessIsMore (Ours) | **92%** | 98% | 98% | **100%** | **100%** |
>
> **Table 14 Caption:** Results of the 10k-context Needle-in-the-Haystack test on the MHA-based LongChat-7B-v1.5-32k model. LessIsMore achieves equal or superior accuracy compared to selection-based baselines such as Quest[1] and TidalDecode[2], and substantially outperforming eviction-based approaches including H2O[3], TOVA[4], and StreamingLLM[5].
>
> > **W2:** Some gains may be tied to custom GQA kernel support. Can you quantify benefits under other stacks like (vLLM/Flash-Attention) and with speculative decoding or KV-cache quantitation?
>
> **A2**: We thank the reviewer for mentioning the importance of the actual benefit on stacks like vLLM/SGLang/FlashAttention; our kernels are indeed implemented on top of the FlashInfer kernel library, which is the state-of-the-art attention kernel library used by vLLM/SGLang. In addition, for our efficiency evaluation, our baseline also used the optimal FlashInfer library to make it a fair comparison. We have included additional numbers that evaluate the end-to-end speed-ups (up to **1.5x**) on top of the SGLang serving stack shown below (also in paper's **Table 8**).
>
> Our technique is essentially orthogonal to speculative decoding and KV-Cache quantization because our methods reduce both the memory transfer of KVs from global to shared memory and the compute, so we can expect a similar amount of kernel speed-up.
>
> | Method | 16K | 32K | 64K |
> |---|---|---|---|
> | SGLang + LessIsMore-2K | 1.11 | 1.25 | 1.51 |
> | SGLang + LessIsMore-4K | 1.09 | 1.22 | 1.48 |
>
> **Table 8 Caption:** End-to-end TBT speed-up of LessIsMore on SGLang serving stack under different context lengths.

---

> ### Author Response · Authors · 2025-11-21
> **Reply to Reviewer K4Av [2/2]**
>
> > **W3:** Is the cross-head overlap is something specialised to GQA or a general property? Can you show comparisons between MHA and GQA by keeping other factors fixed?
>
> **A3**: LessIsMore is designed to perform highly accurate reasoning while reducing the latency in long decoding. However, more than a design specialized and effective for GQA-only models, the cross-head overlap property also exhibits other attention mechanisms, such as MHA. We have added new experiments showing the exceptional long-context capabilities of LessIsMore on the non-reasoning MHA-based model LongChat in **Table 14** (also attached above). LessIsMore significantly outperforms other sparse attention baselines such as H2O, StreamingLLM, and TOVA and achieves similar accuracies to Quest and TidalDecode on the 10K-needle-in-the-haystack task. This further validates the generalizability of LessIsMore on tasks and attention mechanisms beyond reasoning and long-context.
>
>
>
> References:
>
> [1] J Tang, et al. "Quest: Query-Aware Sparsity for Efficient Long-Context LLM Inference." arXiv preprint arXiv:2406.10774, 2024.
>
> [2] L Yang et al. "TidalDecode: Fast and Accurate LLM Decoding with Position Persistent Sparse Attention." arXiv preprint arXiv:2410.05076, 2024.
>
> [3] Z Zhang, et al. "H₂O: Heavy-Hitter Oracle for Efficient Generative Inference of Large Language Models." arXiv preprint arXiv:2306.14048, 2023.
>
> [4] M Oren, et al. "Transformers are Multi-State RNNs." arXiv preprint arXiv:2401.06104, 2024.
>
> [5] G Xiao, et al. "Efficient Streaming Language Models with Attention Sinks." arXiv preprint arXiv:2309.17453, 2023.

---

### Official Review · Reviewer_WsK9 · 2025-11-02

**Soundness:** 3
**Presentation:** 3
**Contribution:** 2
**Rating:** 6
**Confidence:** 2

**Summary:**

The paper introduces LessIsMore, a training-free sparse attention mechanism tailored for reasoning LLMs (DeepSeek-R1, Qwen3).
It identifies two empirical “locality” phenomena during reasoning: (1) Spatial locality across attention heads: token-importance overlaps heavily across heads within the same layer. (2) Recency locality: recently generated tokens are consistently attended more.


Based on these findings, the authors propose: (a) Cross-Head Unified Sparse Attention (CUSA): aggregate per-head top-k tokens globally into one shared subset. (b) Stable Recency Window: reserve a fixed ratio r (25%) of tokens for the most recent context.
They show Up to 1.6X end-to-end decoding speedup (and 1.72X kernel-level speedup) with no accuracy loss across AIME-24/25, GPQA, and MATH500, compared to TidalDecode and SeerAttention-r.

**Strengths:**

**1) Strong empirical motivation and analysis:** Figure 2 & Appendix A.4 genuinely show cross-head overlap and consistent recency trends.
Verified with visualizations at 10K–25K decoding steps; supports the “shared importance” hypothesis.

They correctly recognize reasoning != retrieval: long decoding amplifies error accumulation (Fig. 1b).

**2) Novel, simple design:** The Cross-Head Unified Selection (Algorithm 1, lines 11–14) is conceptually clean and implementable atop any existing sparse attention

**3) Training-free and kernel-friendly:** No retraining or finetuning is required

**Weaknesses:**

1)  The evaluation uses 16 samples (AIME-24/25) and 8 samples (MATH500/GPQA) to report pass@1 accuracy and compare methods (see §4.1). With such small n, 2–3 pp differences are plausibly within noise; there are no confidence intervals, seed variance, or significance tests.

2) Re-selection and layer choices ad-hoc? Section 4.1 fixes token-selection layers (12, 20) based on a “needle-in-the-haystack” test, but this heuristic is weakly justified and may bias results.


3) Sparse selection applied only at two layers, unclear if additional layers (e.g., alternating pattern) could further speed up.

**Questions:**

1) Layer sensitivity: Why choose Layer 12 and 20 specifically? does earlier or later selection harm performance?

2) Ratio r = 0.25: Is 25% constant across all tasks, or was it tuned? Can adaptive r perform better?

3) Have you empirically measured correlation between attention recall and benchmark accuracy across tasks?

---

> ### Author Response · Authors · 2025-11-21
> **Reply to Reviewer WsK9 [1/2]**
>
> We sincerely thank the reviewer for the valuable feedback.
>
> **TL;DR:** We have increased sample sizes to 64 (AIME) and 16 (GPQA) with extended token budgets up to 8K. Added StreamingLLM baseline across all tasks (Figure 4 in 4.2, Tables 3–4 in A.1.2). LessIsMore consistently achieves best accuracy with minimal variance. Re-selection layers (12, 20) are validated via needle-in-haystack experiments and remain consistent across model families.
>
> > **W1:** The evaluation uses 16 samples (AIME-24/25) and 8 samples (MATH500/GPQA) to report pass@1 accuracy and compare methods (see §4.1). With such small n, 2–3 pp differences are plausibly within noise; there are no confidence intervals, seed variance, or significance tests.
>
> **A1**: We thank the reviewer for asking about sample size. We want first to clarify that the sample size number here is not how many problems/requests we have evaluated from the corresponding dataset (AIME-24/25, GPQA-Diamond/MATH500). We indeed evaluate all the problems in each dataset, which is 30 problems for AIME-24/25, 198 problems in GPQA-Diamond and 500 problems for MATH500. But since the model generation is commonly stochastic, for each problem we generate more than one complete trace with an answer, which we refer to as the sample size. Then we average over all the generated responses to get the averaged accuracy for such a problem. We apologize for any confusion that we introduced by overusing the sample size term, and we have refined our writing accordingly. Under such cases, a sample size of 16 is typically low-variance in **Table 5** (attached below), showing relationships of accuracy variance vs. the number of answers sampled per problem on different tasks. But to further address the reviewer's concern, we increased the sample size to 64 for AIME-24 and AIME-25 and 16 for GPQA-Diamond. Final accuracies and differences between new and old ones are shown in **Table 3 & 4**. We have plotted a new figure to include larger token budgets and a new baseline StreamingLLM[3] in **Figure 4**. For all the sample sizes used in our paper and updated ones, LessIsMore always achieves the best accuracy and small variance across all token budgets and achieves a similar accuracy with the full attention baseline with small token budgets.
>
> | Model (Task) | Task / Sample Size | 8 | 16 | 64 |
> |---|---|---|---|---|
> | Qwen-3-8B (Variance) | AIME-24 | ±1.54 | ±0.98 | ±0.56 |
> | | AIME-25 | ±1.76 | ±1.12 | ±0.58 |
> | | MATH500 | ±0.14 | ±0.11 | ±0.05 |
> | | GPQA | ±0.59 | ±0.43 | ±0.20 |
>
> **Table 5 Caption:** The sampled variance of AIME-24/25, MATH500, and GPQA stay minimal (<0.6) with 64, 8, and 16 samples per problem.
>
> > **W2:** Re-selection and layer choices ad-hoc? Section 4.1 fixes token-selection layers (12, 20) based on a "needle-in-the-haystack" test, but this heuristic is weakly justified and may bias results.
>
> **A2**: The choice of re-selection layer is not random but justified in A.3. More specifically, we decide the best selection layer using a simple retrieval needle-in-the-haystack (NiTH) task and apply the same re-selection layer for ALL tasks of the same model. Earlier work TidalDecode [2] has studied the importance of the re-selection layer and shows that across different models within the same model family, optimal re-selection layer choices tend to be consistent. This consistency in optimal re-selection layers is also supported by our new **Table 11** in A.4 summarizing re-selection layer used in paper and evaluation results on NiTH by applying LessIsMore to the Llama3- and Llama-3.1-8B model in **Table 6** (attached below), where Layer 13 is used as the re-selection layer for both models to achieve full accuracy on NiTH task with up to 99.9% sparsity.
>
> | Model (context length) | Method / Budget | K=32 | K=64 | K=128 | K=256 | K=512 |
> |---|---|---|---|---|---|---|
> | Llama-3-8B (10K) | Quest | 74% | 84% | 99% | 98% | **100%** |
> | | TidalDecode | 88% | 98% | **100%** | **100%** | **100%** |
> | | LessIsMore (Ours) | **100%** | **100%** | **100%** | **100%** | **100%** |
> | Llama-3-8B (100K) | Quest | 38% | 50% | 65% | 87% | 98% |
> | | TidalDecode | 86% | 92% | **100%** | **100%** | **100%** |
> | | LessIsMore (Ours) | **98%** | **100%** | **100%** | **100%** | **100%** |
> | Llama-3.1-8B (10K) | Quest | 74% | 86% | 94% | **100%** | 98% |
> | | TidalDecode | **100%** | **100%** | **100%** | **100%** | **100%** |
> | | LessIsMore (Ours) | **100%** | **100%** | **100%** | **100%** | **100%** |
> | Llama-3.1-8B (32K) | Quest | 78% | 88% | 92% | **100%** | **100%** |
> | | TidalDecode | **98%** | **100%** | **100%** | **100%** | **100%** |
> | | LessIsMore (Ours) | **100%** | **100%** | **100%** | **100%** | **100%** |
>
> **Table 6 Caption:** Results of 10K-, 32K-, and 100K-context NiTH on non-reasoning models Llama-3-8B-Instruct-Gradient-1048k and Llama-3.1-8B-Instruct using the PG-19-mini dataset.

---

> > ### Author Response · Authors · 2025-11-21
> > **Attached Table 11: Summary of Re-selection Layers**
> >
> > | Model | Family | Attn Type | Reasoning | Re-selection Layer |
> > |---|---|---|---|---|
> > | LongChat-7B-v1.5-32k | Llama2 | MHA | ✗ | 7 |
> > | Llama-3-8B | Llama3 | GQA | ✗ | 13 |
> > | Llama-3.1-8B | Llama3 | GQA | ✗ | 13 |
> > | DeepSeek-R1-Distill-Llama-8B | Llama3 | GQA | ✓ | 12 |
> > | Qwen3-4B | Qwen3 | GQA | ✓ | 20 |
> > | Qwen3-8B | Qwen3 | GQA | ✓ | 12 |
> > | Qwen3-14B | Qwen3 | GQA | ✓ | 12 |
> >
> > **Table 11 Caption:** Re-selection layers used for evaluations, with model family and task type annotations.

---

> ### Author Response · Authors · 2025-11-21
> **Reply to Reviewer WsK9 [2/2]**
>
> > **W3:** Sparse selection applied only at two layers, unclear if additional layers (e.g., alternating pattern) could further speed up.
>
> **A3**: We thank the reviewer for mentioning the layer composition of LessIsMore. In our current design, there are 2 full attention layers and 2 selection layers, while all the rest layers are sparse layers, as illustrated in Algo. 1. For example, for Qwen3-8B with 36 layers, there are 32 sparse attention layers, which lead to sparsity up to 90%, a higher sparsity ratio compared with the interleaved model architecture as in [1]. The philosophy behind this design is to maintain the accuracy as high as a dense model while performing as few computations as possible. In this case, LessIsMore keeps the fewest possible number of selection layers, which is two in all the models we tested, to achieve the optimal efficiency by up to 1.6x speedup without hurting the accuracy compared to the full attention baseline shown in Fig.5(a) in 4.4 and Fig.4 in 4.2 . Adding more selection layers will not significantly improve the accuracy, given that we are already close to the dense model's performance, but will reduce the speedup. Moreover, we have added new experiments in **Table 12** (attached below) showing the catastrophic effect on accuracy of removing or changing the reselection layer to justify the optimal balance of efficiency and accuracy of our current design.
>
> | Model (Task) | Method / Budget | K=2000 | K=4000 | K=6000 | K=8000 |
> |---|---|---|---|---|---|
> | Qwen-3-8B (AIME-24) | LessIsMore+(None) | 58.02 | 63.33 | 66.17 | 70.00 |
> | | LessIsMore+L5 | 53.33 | 62.60 | 70.31 | 74.68 |
> | | LessIsMore+L18 | 71.67 | 72.91 | 73.33 | 74.79 |
> | | LessIsMore+L30 | 63.23 | 65.83 | 70.10 | 75.27 |
> | | **LessIsMore+L12 (Ours)** | **73.00** | **75.56** | **76.45** | **76.67** |
> | | Full Attention | 74.48 | 74.48 | 74.48 | 74.48 |
>
> **Table 12 Caption:** Results of 2K-, 4K-, 6K-, and 8K-token-budget with sampling 64 answers per problem in AIME-24 evaluated on Qwen3-8B. LessIsMore+Lx represents using Layer x as re-selection layer. Full Attention is shown for reference.
>
> > **Q1:** Layer sensitivity: Why choose Layer 12 and 20 specifically? does earlier or later selection harm performance?
>
> **A4**: We agree that the re-selection layer is essential to accuracy. The re-selection layer plays a critical role in accuracy and is determined by a one-time simple needle-in-the-haystack experiment. Prior works [2] have found that within the same model family, models tend to have a similar pool of optimal layers. Our observation that choosing layer 12 and 20 as re-selection layers provides the best accuracy on the needle-in-the-haystack task and thus applied in LessIsMore for all reasoning tasks supports the claim with updated experiments in Table 3 and 4. We also added the experiments with earlier, close, and later re-selection layers on AIME-24 for Qwen3-8B model in **Table 12** (also attached above). Notice that LessIsMore maintains the same re-selection layer for the same model across ALL evaluations in Table 1, 2, 3, and 4.
>
> > **Q2:** Ratio r = 0.25: Is 25% constant across all tasks, or was it tuned? Can adaptive r perform better?
>
> **A5**: We thank the reviewer for asking the recency window. The recency window ratio here is a hyper-parameter, and we indeed conducted comprehensive ablation studies as shown in Figure 7 in 4.4 for it. We observed that a static 25% for the recency window achieves optimal performance across all models, token budgets, and reasoning tasks, which indicates that the recency window ratio is insensitive to model and evaluation task types. Our ablation study also shows that a 25% ratio provides the highest attention recall throughout the generation and thus leads to the correct output. Moreover, heatmaps in Figure 9 in A.6 in demonstrate that regardless of different tasks, decoding lengths, and token budgets, the identified range of recency window/token budget stays consistent, close to 25%. This is the first comprehensive analysis of the effect of the most recent tokens on reasoning models.
>
> > **Q3:** Have you empirically measured correlation between attention recall and benchmark accuracy across tasks?
>
> **A6**: We thank the reviewer for asking the relationship between attention recall and accuracy. Yes, Figure 1 in 2.1 measures the discrepancy of attention recall of different sparse attention baselines. We added experiments of a new baseline StreamingLLM [3] and extended the token budgets and larger sample sizes across all tasks in Figure 4 in 4.2. It reveals that attention recall is closely related to accuracy. For example, LessIsMore has much higher attention recall in Figure 1(a) than StreamingLLM; thus, StreamingLLM suffers significant accuracy degradation on all reasoning tasks, even with large token budgets.

---

> ### Author Response · Authors · 2025-11-21
>
> References:
>
> [1]. Agarwal, Sandhini, et al. "gpt-oss-120b & gpt-oss-20b model card." arXiv preprint arXiv:2508.10925 (2025).
>
> [2]. Yang, Lijie, et al. "Tidaldecode: Fast and accurate llm decoding with position persistent sparse attention." arXiv preprint arXiv:2410.05076 (2024).
>
> [3]. Xiao, Guangxuan, et al. "Efficient streaming language models with attention sinks." arXiv preprint arXiv:2309.17453 (2023).

---

### Author Response · Authors · 2025-11-21
**Updated Paper and Additional Experiments over Discussion Period**

We thank the reviewers for their valuable feedback. We are excited and grateful that reviewers recognize LessIsMore as a simple and novel design with strong motivation supported by efficiency kernel-friendly implementations and comprehensive evaluations. We have considered their suggestions and questions seriously and have updated our paper with changes being highlighted. Please see individual replies for details.

## New Experiments in Updated Paper

### Main Reasoning Evaluation
- **Revised Table 1 & 2 + Figure 4**: Extended token budgets to 6K on GPQA-Diamond/MATH500 and 8K on AIME-24/25; added StreamingLLM (attention_sink=8) as baseline
- **New Table 3 & 4**: Evaluated on larger sample sizes (64 for AIME-24/25, 16 for GPQA-Diamond) with differences from old results shown
- **New Table 5**: Variance analysis for each task across different sample sizes

### Long-Context Task Evaluation
- **New Table 6**: Needle-in-the-haystack evaluation results (10K/32K/100K contexts)
- **New Table 7**: LongBench dataset evaluation results (MultiFieldQA, Qasper, HotpotQA, TriviaQA, PassageRetrieval-en)

### Efficiency Evaluation
- **New Table 8**: End-to-end time-between-token speedup of LessIsMore on SGLang
- **New Table 9**: End-to-end single-step decoding latency comparison across all baselines
- **New Table 10**: Kernel-level FLOP count, shared memory usage, and on-device memory analysis

### Ablation and Generalization Studies
- **New Table 11**: Summary of re-selection layers used across models in our paper
- **New Table 12**: Impact evaluation of earlier or later re-selection layers in LessIsMore
- **New Table 13**: Generalization results on MHA-based models (LongChat) with more sparse attention baselines from [1]

## Revisions to Paper (highlighted in red)

**Main Text Updates**:
- Moved ablation study on effectiveness of LessIsMore aggregation in GQA to main text
- Moved analysis of recency window ratio effect to main text
- Revised wording for Figure 2 explanation and experiment section clarifying how we sample answers from each dataset

**New Appendix Sections**:
1. Larger sample size reasoning evaluation
2. Long-context evaluation details
3. Additional end-to-end efficiency analysis
4. Evaluation on MHA-based models

All changes have been highlighted in red in the revised manuscript for easy reference.

[1] Efficient Attention Mechanisms for Large Language Models: A Survey, arXiv:2507.19595

---

### Author Response · Authors · 2025-12-03
**Summary for Area Chair over Discussion Period [1/2]**

We sincerely thank reviewers for their constructive feedback. All reviewers recognize LessIsMore as a **simple and novel design** with **strong empirical motivation**, **kernel-friendly implementation**, and **comprehensive evaluations**. Reviewer Gm7y, the only reviewer who replied over the discussion period, has admitted that our updated experiments already resolved all of his/her concerns. Below, we categorize the main concerns raised by all reviewers and summarize how we have addressed them.


---


## Main Concerns & Responses


### 1. Statistical Robustness & Sample Size Clarification
**Raised by:** WsK9 (W1), nhSJ (W3)


**Concern:** Evaluation uses 16/8 samples, which might be too small to ensure robust outputs.


**Clarification & Response:**
We have clarified that "Sample size" refers to generations per problem, not total problems being evaluated by model. We evaluate **all** problems in all datasets: 30 (AIME-24/25), 198 (GPQA-Diamond), 500 (MATH500). We further increased the generation size per problem to 64 for AIME-24/25, 16 for GPQA-Diamond, and added variance analysis in **Table 5** (attached below), which shows variance stays minimal (<0.6) across our evaluated samples sizes.
We have updated **Results in Tables 3–4** and replotted Figure 4 with new baseline StreamingLLM and extended token budgets, where LessIsMore still consistently achieves best accuracy with small variance across all token budgets.


| Task | Variance @8 samples | Variance @16 samples | Variance @64 samples |
|--------|--------|--------|--------|
| AIME-24 | ±1.54 | ±0.98 | ±0.56 |
| AIME-25 | ±1.76 | ±1.12 | ±0.58 |
| MATH500 | ±0.14 | ±0.11 | ±0.05 |
| GPQA | ±0.59 | ±0.43 | ±0.20 |
Table 5 Caption: The sampled variance of AIME-24/25, MATH500, and GPQA stay minimal (<0.6) with 64, 8, and 16 samples per problem.


---


### 2. More Sparse Attention Baselines
**Raised by:** WsK9, nhSJ (W6), Gm7y (W2)


**Concern:** Missing StreamingLLM baseline; need extended token budgets beyond 6K for AIME-24/25 and 4K for GPQA/MATH .


**Response:** We have added **StreamingLLM** (attention_sink=8) across all reasoning tasks and extended token budgets to **8K for AIME-24/25**, **6K for MATH500/GPQA**. In updated **Figure 4**, StreamingLLM suffers significant accuracy degradation due to the "lost in the middle" problem—it discards middle tokens. LessIsMore explicitly addresses this via middle token selection (75% budget) + recency window (25% budget). We further added comparisons with **H2O, TOVA** on long-context tasks (**Table 14**), where LessIsMore significantly outperforms them.


---


### 3. Model Generalization Beyond GQA
**Raised by:** K4Av (W1, W3), nhSJ (W2)


**Concern:** Results focus on Qwen-3 and GQA mechanism; unclear if method generalizes to other models like Llama-3.x or MHA architectures.


**Response:**
In the original paper, we had already presented reasoning evals of DeepSeek-R1-Distill-Llama-8B, a Llama3-based model, in **Figure 4**. We further extended with the StreamingLLM baseline and larger token budgets (**Tables 1–4**). To validate generalization of LessIsMore on MHA, evaluations on **LongChat-7b-v1.5-32k (MHA)** for needle-in-the-haystack have been added to **Table 14**, where LessIsMore achieves 92% accuracy at K=32 on MHA model, significantly outperforming H2O (0%), TOVA (0%), StreamingLLM (1%). Please refer to Reply to Reviewer K4Av [1/2] and [2/2] for details on the generalization of LessIsMore on different attention mechanisms.


---


### 4. Long-Context Task Evaluation
**Raised by:** nhSJ (W5, W7), Gm7y (Q2)


**Concern:** Method targets reasoning but lacks evaluation on broader long-context tasks (LongBench, needle-in-a-haystack).


**Response:** we have added long-context benchmark evaluation as follows:
- **Needle-in-the-Haystack** (**Table 6 and 14**): Evaluated at 10K/32K/100K contexts on LongChat-2-7B, Llama-3-8B, and Llama-3.1-8B
 - LessIsMore achieves **100% accuracy** at K=32 for 10K context (vs. Quest 74%, TidalDecode 88%)
 - At 100K context: LessIsMore 98%→100% (K=32→64) vs. Quest 38%→50%
- **LongBench** (**Table 7**): 5 datasets testing retrieval, multi-hop QA, multi-document comprehension. LessIsMore (K=4096) achieves **44.78 avg score** vs. Full Attention 44.08, TidalDecode 44.56, Quest 42.62, and matches or exceeds full attention while using sparse budget across all tasks


---

---

> ### Author Response · Authors · 2025-12-03
> **Summary for Area Chair over Discussion Period [2/2]**
>
> ---
>
>
> ### 5. Efficiency Analysis & Memory Footprint
> **Raised by:** nhSJ (W4, Q3), Gm7y (W1, Q1), K4Av (W2)
>
>
> **Concern:** Need comprehensive efficiency metrics (FLOPs, latency, memory) and end-to-end measurements on real serving stacks.
>
>
> **Response:**
> - **Kernel-level analysis** (**Table 10**): FLOPs, G2S transfer, memory, latency at 2K budget / 16K context (data transfer, memory, latency: the lower the better)
>
>
> | Method | FLOPs | G2S Transfer | Memory | Latency |
> |--------|-------|--------------|--------|---------|
> | LessIsMore | 1.05M | 1.04MB | 8.38MB | 20.08 μs |
> | TidalDecode | 1.05M | 2.34MB | 8.38MB | 32.12 μs |
> | Quest | 1.05M | 2.34MB | 8.38MB | 32.12 μs |
> | Full Attention | 8.40M | 8.38MB | 8.38MB | 76.38 μs |
>
> Table 10 Caption: Kernel-level FLOP count, global-to-shared (G2S) memory transfer, on-device memory consumption, and per-kernel latency for sparse attention with a 2K token budget and 16K context length on DeepSeek-R1-Distilled-LLaMA-8B.
>
> - **End-to-end latency** (**Table 9**): SGLang + FlashInfer stack integration, single-step decoding, validating practical applicability with our kernel implementation
>  - LessIsMore: 23.0ms (16K) → 24.1ms (64K)
>  - Full Attention: 25.3ms (16K) → 34.4ms (64K)
> - **End-to-end speedup** (**Table 8**): Up to **1.51× TBT speedup** at 64K context. Please refer to Reply to Reviewer Gm7y [1/3] and [2/3] for a detailed comparison with other baselines.
> - Our approach is also orthogonal to speculative decoding and KV-cache quantization
>
> ---
>
> ### 6. Design Choices: Re-selection Layers & Recency Window
> **Raised by:** WsK9 (W2, W3, Q1, Q2), K4Av
>
>
> **Concern:** Re-selection layer choices (12, 20) and 25% recency ratio appear ad-hoc and weakly justified.
>
>
> **Response:**
>
>
> **Re-selection layers** are determined by a one-time needle-in-the-haystack experiment for each model. The same layer is used for ALL tasks of the same model, as shown in **Table 11**. Prior work (TidalDecode) shows that optimal layers are consistent within model families, supported by optimal layers in **Table 11**. Further evaluation shows catastrophic accuracy drop with earlier, later, or no re-selection layer choice in **Table 12**.
>
>
> **Recency Window (r=0.25):**
> The ratio is not tuned per task. We provided a comprehensive ablation in **Figure 7** of the original paper, showing 25% is optimal across different ratios. Moreover, heatmaps from **Figures 8–9** demonstrate consistency regardless of decoding length or task. To our bset knowledge, this is the first work to identify and validate this constant ratio for reasoning models.

---

### Meta-Review · Area_Chair_M6Qk · 2026-01-07

**Summary:**

The main concerns of the reviewers can be summarized as follows:
- **Generalizability beyond the considered setup.** Reviewers have requested applying the proposed method on other model architectures (that utilize MHA) and benchmarks.
- **Less justified design choices.** It was not very clear why the proposed method of recency window size, index sharing, re-selection and layer choice is appropriate for the final objective.
- **Robustness of the evaluation.** There were some concerns and questions regarding the statistical reliability, lacking baselines, and lacking efficiency metrics.

While many of the concerns have been addressed well by the reviewers, several concerns remain. First, the evaluation has been conducted on essentially two models on useful benchmarks---needle-in-the-haystack is a very narrow benchmark. Also, even after reading the current version of the manuscript and the rebuttal carefully, it was very difficult for the AC to clearly understand why the proposed design choices should be that way---it is suggested that the authors reorganize the explanations, in order to improve the clarity. Finally, the conceptual novelty of the scheme proposed in this paper was not very outstanding (or at least very difficult for me to catch). For these reasons, I recommend rejection. It was, however, a close call.

**Reviewer Concerns:**

- **Generalizability beyond the considered setup.** This was partially resolved, but the evidences on applicability to wide range of architectures is still the fully convincing.
- **Less justified design choices.** Even after reading all explanations, these are still not straightforward.
- **Robustness of the evaluation.** This point has been resolved well.

**Reviewer Scores:**

It is not very likely that both k4av and nhsj---the negative reviewers---would have raised score. This is especially so for k4av, who had concerns with the number of models considered. Even if both reviewers raised the score, it would have been a borderline score.

---

### Decision · Program_Chairs · 2026-01-26

Reject